# Generalize Learned Heuristics to Solve Large-scale Vehicle Routing Problems in Real-time

**Qingchun Hou, Jingwei Yang, Yiqiang Su, Xiaoqing Wang, Yuming Deng**
Alibaba Group
`houqingchun.hqc@alibaba-inc.com, hqc16@tsinghua.org.cn`

## Abstract

Large-scale Vehicle Routing Problems (VRPs) are widely used in logistics, transportation, supply chain, and robotic systems. Recently, data-driven VRP heuristics are proposed to generate real-time VRP solutions with up to 100 nodes. Despite this progress, current heuristics for large-scale VRPs still face three major challenges: 1) Difficulty in generalizing the heuristics learned on small-scale VRPs to large-scale VRPs without retraining; 2) Challenge in generating real-time solutions for large-scale VRPs; 3) Difficulty in embedding global constraints into learned heuristics. We contribute in the three directions: We propose a Two-stage Divide Method (TAM) to generate sub-route sequence rather than node sequence for generalizing the heuristics learned on small-scale VRPs to solve large-scale VRPs in real-time. A two-step reinforcement learning method with new reward and padding techniques is proposed to train our TAM. A global mask function is proposed to keep the global constraints satisfied when dividing a large-scale VRP into several small-scale Traveling Salesman Problems (TSPs). As result, we can solve the small-scale TSPs in parallel quickly. The experiments on synthetic and real-world large-scale VRPs show our method could generalize the learned heuristics trained on datasets of VRP 100 to solve VRPs with over 5000 nodes in real-time while keeping the solution quality better than data-driven heuristics and competitive with traditional heuristics.

## 1 Introduction

Vehicle Routing Problems (VRPs) are widely used in logistics, supply chain, transportation, and robotic systems (Toth & Vigo, 2002b; Golden et al., 2008; Bullo et al., 2011). For instance, on e-commerce platforms, hundreds and thousands of goods are sold in real-time and then transported to customers with maximum efficiencies, minimum number of vehicles, and shortest distance. Therefore, more large-scale VRPs need to be solved in real-time to improve logistics or transportation efficiency (Dong et al., 2021; Duan et al., 2020). Although VRP is one of the most well-studied combinatorial optimization problems, the large-scale VRP is still challenging due to its NP-hard characteristic (Golden et al., 2008). Exact methods or solvers (such as branch and bound (Toth & Vigo, 2002a), branch and cut (Naddef & Rinaldi, 2002), column generation (Chabrier, 2006), Gurobi, and Cplex) could obtain global optimal solutions on small-scale VRPs with theory guarantee. However, these methods are time-consuming and hard to be extended to large-scale VRPs because permutation number is growing exponentially. Traditional heuristics or solvers could solve small-scale VRPs quickly with near-optimal solutions. Some heuristics could be extended to solve large-scale VRPs (Ortools (Perron & Furnon), LKH3 (Helsgaun, 2017), HGS (Vidal, 2022; Vidal et al., 2012), and SISRs (Christiaens & Vanden Berghe, 2020)). However, massive iterations are needed to obtain good solutions. The algorithm for solving large-scale VRP in real-time (**seconds**) still lags behind.

Recently, data-driven methods are proposed to learn heuristics for constructing VRP solutions directly (Vinyals et al., 2015). Deep learning methods like Transformer (Vaswani et al., 2017; Kool et al., 2019; Peng et al., 2020) and Graph neural network (Kipf & Welling, 2016; Khalil et al., 2017; Joshi et al., 2019) are used to extract hidden states of VRPs and TSPs, which is called Encoder. The solution sequences of VRPs are then generated in an autoregressive way from the hidden states,

which is called Decoder. Reinforcement learning techniques are also applied to train the encoder-decoder model (sequence-to-sequence model) to improve its accuracy (Nazari et al., 2018). These learn-to-construct heuristics can outperform or be comparable to traditional VRP heuristics with up to 100 nodes. However, when it comes to large-scale VRPs (over 1000 nodes), the learned heuristics still face three challenges: 1) the training of data-driven large-scale VRP model is time-consuming and computationally expensive. For instance, the computation complexity and memory space of training the Transformer are quadratic to the lengths of the input sequence (nodes number of VRP) (Kool et al., 2019; Kitaev et al., 2019); 2) the model trained on small-scale VRPs is difficult to be generalized to large-scale VRPs because the nodes distribution of large-scale VRPs in test dataset is different from that of the small-scale VRPs in the training dataset; 3) the global constraints like maximum vehicle number are hard to be encoded in the encoder-decoder model because global constraints become active only at the end of the sequence.

Although the limitations of traditional and data-driven methods, we ask: *Could we generalize the learned heuristics to solve large-scale VRPs in real-time by taking advantages of both data-driven and traditional methods?* We try to answer this question from the following perspectives:

**1)** Although the traditional heuristic methods are time-consuming when solving large-scale VRPs, they can quickly obtain optimal or near-optimal solutions with some theory guarantees when solving small-scale VRPs. We observe that vehicle capacity for real-world large-scale VRPs is limited, and each vehicle serves a few customers. If we know the customers that each vehicle needs to serve, then the original large-scale VRP could be divided into several small-scale TSPs, which could be solved by traditional heuristics quickly and parallelly.

**2)** The generalization of data-driven heuristics to large-scale VRPs is difficult because the sequence-to-sequence model needs to learn the distribution of each node in the long sequence. We observe that if we just model the distribution of sub-routes and ignore the order of nodes inside a sub-route, then we could possibly better generalize the model trained on small-scale VRPs to solve large-scale VRPs.

**3)** Although the global constraints are only active at the end of the sequence, we could design a global mask function with theory guarantee to prevent the infeasible solution beforehand. In addition, the global constraints could include some prior information, which helps improve the generalization of the learned heuristics. For instance, we observe that the predefined maximum vehicle number could provide some global information about the possible range of the optimal vehicle number in the testing dataset, which could help identify the minimum travel length.

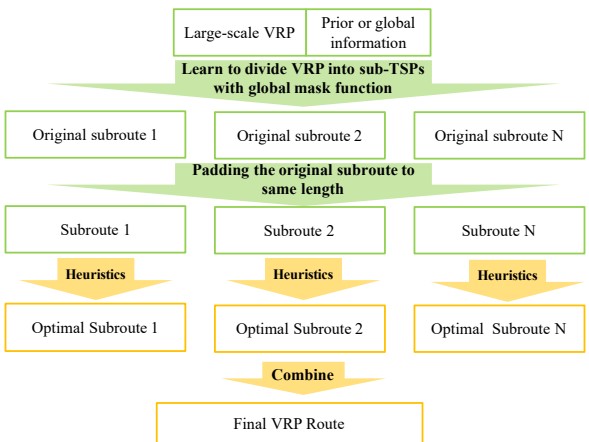

Figure 1: **Our TAM framework**. In the first stage (green), a learned model divides large-scale VRP into several small TSPs while satisfying VRP constraints like capacity and maximum vehicle number. Then, the original TSPs are padded to the same number of nodes at the training time. In the second stage (orange), all TSPs or sub-routes with simple constraints are optimized in parallel.

Driven by the above analysis, we present a Two-stage Divide Method (TAM) in Figure 1 for generalizing the learned heuristics to solve large-scale VRPs in real-time with a zero-shot way. Our TAM combines the real-time advantages of data-driven methods and the generalization advantages of traditional heuristics. It first divides a large-scale VRP into several smaller sub-routes by generating

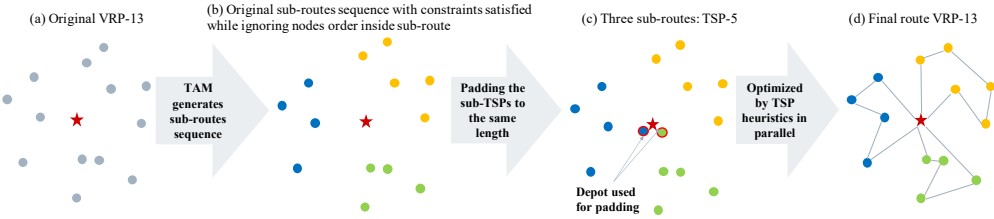

Figure 2: Illustration of dividing a VRP 13 into three TSP-5s, which are optimized in parallel.

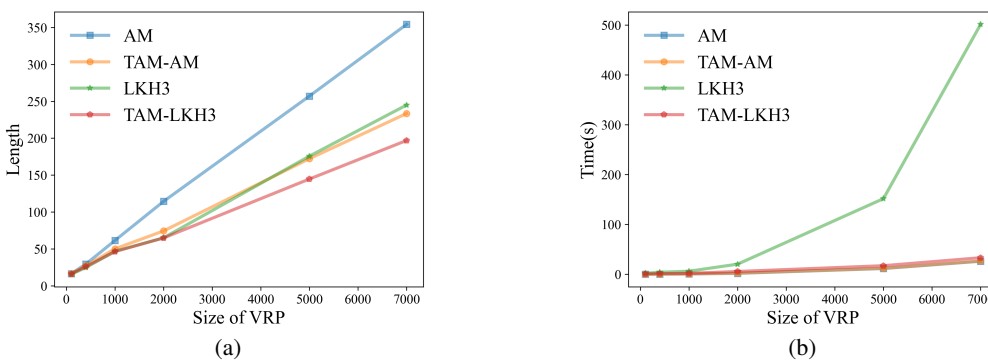

Figure 3: The performance of TAM and benchmark methods agaist the size of VRP (a) Route length (b) Solving time. **TAM-LKH3:** Our TAM with LKH3 as the second-stage solver; **TAM-AM:** Our TAM with Attention Model as the second-stage solver; **AM:** Attention Model; **LKH3:** LKH3 solver.

sub-route sequence using a learned encoder-decoder model. The encoder-decoder model is trained by a two-step Reinforcement Learning (RL) with a novel reward that encodes the distribution of sub-route sequence. To accelerate the training process, we propose a padding technique to pad the original TSPs to the same number of nodes at the training time. A global mask function is also proposed to encode global constraints like maximum vehicle number in this stage. In the second stage, all small-scale sub-routes could be solved in parallel using traditional heuristics or learned heuristics. To better understand our method, we illustrate the process by splitting a VRP 13 into three smaller TSP-5s and obtaining the final route by combining the three optimized TSPs in Figure 2.

Technically, we summarize our key **contributions** as follows:

**(1)** We propose a new formulation and Two-stage Dividing Method for generalizing the learned heuristics trained on small-scale VRPs to solve large-scale VRPs in a real-time and zero-shot way.

**(2)** We propose three techniques to improve the zero-shot generalization ability of our TAM: 1) Propose **generating sub-route sequence** to take advantages of both learned and traditional heuristics; 2) Propose **a two-step RL training method with new reward and padding method** to accelerate training process and make the TAM invariant to node sequence of sub-route; 3) Propose a **global mask function** to encode maximum vehicle number constraint with theory guarantee. A proof of the mask function and a generalization analysis of TAM are provided in Section A.4 of Appendix.

**(3)** We validate our TAM on synthetic and real-world large-scale VRPs. The results show our TAM could generalize the learned heuristics trained on VRP 100 to solve VRPs with over 1000 nodes in real-time while keeping the solution quality competitive with traditional heuristics. For VRP 2000, the solution quality of our TAM is over 50% better than the data-driven Attention Model. Example results of generalizing to VRP 2000 and real-world VRP 1040 are shown in Figure 5 of Appendix.

**(4)** Our TAM can scale to VRPs with over 5000 nodes. For VRP 7000, the solution quality of our TAM is about 20% better than LKH3 with only 6.6% solution time. The scalability performance of our TAM (TAM-LKH3 and TAM-AM) against the size of VRP is shown in Figure 3. Example results of generalizing to VRP 7000 and 5000 are shown in Figure 7 and 8 of Appendix.

## 2 RELATED WORK

As above mentioned, VRPs are usually solved by exact methods, traditional heuristics, and data-driven methods. The exact methods are time-consuming and incapable of solving large-scale VRPs in a reasonable time. Therefore, we mainly focus on the traditional heuristics and learned methods. In the following part, we summarize the related work about the learned methods. More comprehensive reviews about the traditional heuristics and learned methods could be found in Appendix A.5.

**Learned methods** include learn-to-construct method and learn-to search method. **The learn-to-construct method** constructs good VRP solutions in real-time directly without iterations. Vinyals et al. first proposed using pointer network (PN) to generate TSP solution in real-time (Vinyals et al., 2015). From there, several improvements are witnessed (Nazari et al., 2018; Kool et al., 2019; 2021). The learn-to-construct ideas are also applied in other variant VRPs and get promising results (Delarue et al., 2020; Peng et al., 2020; Falkner & Schmidt-Thieme, 2020; Xin et al., 2020). However, due to the difficulty of training the model on large-scale VRPs, generating solutions for VRPs with over 400 nodes is still challenging (Fu et al., 2021; Joshi et al., 2020; Ma et al., 2019). **Different from previous works**, our TAM generalizes the learned heuristics to solve VRPs with over 1000 nodes in real-time by taking the advantages of traditional heuristics, data-driven heuristics, and prior information. To the best of our knowledge, our TAM is the first learn-to-construct method that could solve VRPs with over 5000 nodes in seconds with zero-shot generalization. In contrast, **the learn-to-search method** is mainly used in exact methods (Gasse et al., 2019) and traditional heuristics (Lu et al., 2019; Khalil et al., 2017; Chen & Tian, 2019; Hottung & Tierney, 2019; Xin et al., 2021; Chen et al., 2020), such as learning better large neighborhood search heuristics (Hottung & Tierney, 2019; Chen et al., 2020), designing better constructors, destructors, and improvement operators (Lu et al., 2019; Khalil et al., 2017) to accelerate the search process. However, the iterative search is still necessary.

**Decomposition technique** has been used in traditional heuristics (Queiroga et al., 2021; Zhang et al., 2021; Bosman & Poutré, 2006; Ventresca et al., 2013; Alvim & Taillard, 2013; Lalla-Ruiz & Voss, 2016; Taillard & Helsgaun, 2019). Recently, the decomposition ideas are introduced to data-driven heuristics for solving TSP and VRP (Fu et al., 2021), such as divide and conquer networks for TSP (Nowak et al., 2018) and learn-to-delegate method for VRP(Li et al., 2021). They both belong to learn-to-search methods. **Different from previous works**, our TAM contributes to generalize learn-to-construct heuristics in real-time, which decomposes VRP as independent sub-routes and just call sub-problem solver once in parallel. Besides, our work could easily encode other VRP constraints by changing mask functions, which is more difficult for the learn-to-search method. In the following sections, we will focus on the learn-to-construct method.

## 3 PRELIMINARY WORK

**VPR definition** VRPs are usually defined as serving all customers' demands from a depot with user-defined objectives while considering some constraints. In this paper, we investigate the CVRP (capacitated vehicle routing problem) with maximum vehicle number constraints $l_m$ while minimizing travel length. Assuming CVRP with $n$ customers, the inputs are points set $\Omega = \{I_0, I_1, \ldots, I_n\}$, where $I_0$ represents depot; $I_i$ represents $i^{th}$ customer. Each customer $I_i = (x_i, y_i, d_i)$ contains three features: x-coordinate $x_i$, y-coordinate $y_i$, and demand of customer $d_i$. The demand that each vehicle serves is constrained by the vehicle's capacity $C$. The number of available vehicles is limited by $l_m$. Our objective is to find an optimal permutation $\boldsymbol{\pi} = \{\pi_0, \pi_1, \ldots, \pi_{n+l}\}$ of $\Omega$ to minimize Euclidean travel length, where $I_0$ is visited $l$ times. A detailed model is shown in Appendix A.1.

**Markov Decision Process Model** Previous learn-to-construct methods such as Attention Model (AM) (Kool et al., 2019) use RL to train a sequence-to-sequence policy for solving the VRPs directly. The corresponding MDP (Markov Decision Process) model is as follows: 1) *State:* The current partial solution of VRP. 2) *Action:* Choose an unvisited customer. The action space is the set of unvisited customers. 3) *Reward:* The negative value of travel distance.

Therefore, the corresponding stochastic policy $p(\pi_t|s_t)$ represents the probability to choose an unvisited customer $\pi_t$ as the next customer, given the current state $s_t$, which contains the information of the instance and current partial solution. Then, the policy for optimal route $\boldsymbol{\pi}$ could be modeled as:

$$p_\alpha(\boldsymbol{\pi}|s) = \prod_{t=1}^{n+l} p_\alpha\left(\pi_t|s_t\right) \tag{1}$$

where $l$ is the vehicle number or sub-route number; $\alpha$ is the parameter of the stochastic policy. In this formulation, the possibility of the next customer should be accurately modeled for each unvisited customer. For large-scale VRPs, generalizing the model to calculate this possibility is much more difficult because of the distribution shift from the training dataset. This is also one of the main reasons behind the generalization failure of current learn-to-construct heuristics.

# 4 METHOD

## 4.1 GENERAL FORMULATION OF TAM

For our TAM in Figure 1, global information is learned in the first stage, while local information like nodes' sequence inside each sub-route is handled in the second stage. Therefore, the most challenging part is learning the best global split while considering complex constraints. To this end, we propose generating sub-route sequence rather than node sequence. The MDP model is designed as follows:

1) *State:* The current partial solution, composed of the chosen sub-routes of VRP.

2) *Action:* Choose an unvisited sub-route. The sub-route for VRP here is defined as **the set of customers** that a vehicle could serve. The action space is the set of unvisited sub-routes.

3) *Reward:* The negative value of **minimal distance** of the chosen sub-route.

Therefore, our policy becomes $p_\theta(r_t|s_t)$, representing the probability to choose an unvisited sub-route $r_t$. The optimal sub-route split $\boldsymbol{r}$ could be modeled as:

$$p_\theta(\boldsymbol{r}|s) = \prod_{t=1}^{l} p_\theta\left(r_t|s_t\right) \tag{2}$$

where $\theta$ is the parameter of the stochastic split policy. In this formulation, we ignore the visiting order of customers (also called nodes) inside each sub-route $r_t$. This configuration is more beneficial for generalizing the learned model to large-scale VRPs because the distribution of sub-routes could be similar between testing dataset and training dataset, although the number and distribution of nodes are different between the two datasets. After generating the optimal sub-route $\boldsymbol{r}$, the optimal route $\boldsymbol{\pi}$ could be modeled as:

$$p(\boldsymbol{\pi}|s) = \left(\prod_{t=1}^{l} p_\theta\left(r_t|s_t\right)\right) \left(\prod_{t=1}^{l} \prod_{i=1}^{l_t} p_\beta\left(\pi_i|r_t, \pi_{t,0:i-1}\right)\right) \tag{3}$$

This formulation means we first obtain the optimal sub-route split and then determine the node sequence in each sub-route independently. The $l_t$ is number of node in the route $r_t$. The $\beta$ denotes the parameters of the policy that determines the customer's order inside a sub-route. The $\pi_{t,0:i-1}$ represents the chosen partial sequence inside sub-route $r_t$. In this paper, we solve the sub-problems using learned or traditional heuristics in parallel. Based on Equation 3, **a generalization analysis** about why our TAM works is shown in Appendix A.4.2.

## 4.2 GENERATING SUB-ROUTE SEQUENCE WITH SEQUENCE-TO-SEQUENCE MODEL

Modeling the policy in Equation 2 faces two challenges. First, the action space is much larger and we need to sample all nodes inside a sub-route while ignoring the nodes' order. Second, mask function for constraints is harder to encode without node sequence (Kool et al., 2019; Nazari et al., 2018). To bridge this gap, we propose to transform the policy in Equation 2 into an equivalent sequence-to-sequence policy (Equation 4) with novel reward to eliminating the effect of nodes' order (Equation 5). We assume the nodes between two depots belong to the same sub-route:

$$p_\theta(\boldsymbol{\pi}|s) = \prod_{t=1}^{n+l} p_\theta\left(\pi_t|s_t\right)$$
$$\pi^0, \ldots, \pi^i, \ldots \pi^l \in \{I_0\}$$
$$\{\pi^{i-1} = \pi_j, \pi_{j+1} \ldots, \pi^i\} \in r_i \quad \forall i = 1, \ldots, l \tag{4}$$
$$r_i \in \Theta_i \quad \forall i = 1, \ldots, l$$
$$l \le l_m$$

where $\pi^i$ is the depot visited in $i^{th}$ time; $r_i$ is $i^{th}$ sub-route; $\Theta_i$ represents the feasible space of $r_i$. The first line of Equation 4 is a sequence-to-sequence policy. The second line shows the times to visit the depot. The third line transforms the sequence of nodes between two depots into the set of nodes (a sub-route problem). The fourth line shows each sub-route should satisfy some local constraints such as capacity constraint. The fifth line shows the global constraint for maximum vehicle number. With this formulation, we could generate sub-route sequence with sequence-to-sequence model. However, the nodes' order inside sub-route still affects the performance of the policy.

## 4.3 NEW REWARD

To make the reward of the chosen sub-route $r_i$ invariant to the order of nodes inside that sub-route, we propose to use the optimal length of the sub-route as its reward $R_i$:

$$R_i = - \min_{\phi} \text{dist}(\phi(r_i)) \quad \forall i = 1, \ldots, l \tag{5}$$

where $\phi()$ is permutation function; dist() is distance function used to calculate route length. Because the optimal length is invariant to the order of inputs, the reward is then invariant to the order of nodes inside the sub-route. The optimization method could be exact methods, traditional or learned heuristics. The accumulated reward $R$ is then used to train our policy (details in Section 4.5):

$$R = \sum_{i=1}^{l} R_i \tag{6}$$

In doing so, we could find the next route while using mask function to satisfy complex constraints.

## 4.4 GLOBAL MASK FUNCTION

For CVRP, each sub-route should satisfy vehicle capacity constraints (local constraints); the whole route should satisfy the predefined maximum vehicle number constraints (global constraints). The global constraints, however, could only be active at the end of a sequence, making it hard to adjust the sequence when the constraints are active and to obtain a feasible solution. Therefore, we propose a new global mask function for global constraints. According to our formulation in Equation 4, the time of visiting depot is equal to the number of vehicles. Therefore, we could constrain the probability to visit depot $p_\theta\left(\pi_t = I_0 | s_t\right)$ beforehand. If the capacity of unused vehicles cannot serve the rest demand, then the current vehicle can not visit the depot:

$$p_\theta\left(\pi_t = I_0 | s_t\right) = \text{softmax}(u_0), u_0 = \begin{cases} \tanh\left(\frac{\mathbf{q}^T \mathbf{k}_0}{\sqrt{dim_\mathbf{k}}}\right) & \text{if } \sum_{i \in \Omega_u} d_i \leq C * (l_m - l_u) \\ -\infty & \text{otherwise.} \end{cases} \tag{7}$$

where $\mathbf{q}$ is the context query; $\mathbf{k}_0$ represents the key of the depot; $\Omega_u$ is the set of unvisited nodes; $l_m$ represents the maximum vehicle number; $l_u$ is the number of used vehicles, including the current vehicle; $dim_\mathbf{k}$ is the dimension of hidden state. The calculation methods of $\mathbf{q}$ and $\mathbf{k}_0$ are shown in Section A.3.3 of Appendix. With Equation 7, we could prevent the violation of the maximum vehicle number constraints by making full use of the current vehicle's capacity.

**Theorem 1** (Global mask function for maximum vehicle number constraint). *Assume that 1) the total capacity of vehicles is larger than the total demand of customers $l_m * C > \sum_i d_i$; 2) the maximum demand of a single customer is much less than the capacity of vehicle $\max_{i \in \{1, \ldots, n\}} d_i \ll C$. Then, the proposed global mask function enforces the satisfaction of the maximum vehicle number constraint $l \leq l_m$.*

**The theoretical proof and remarks of Theorem 1** are provided in Appendix A.4.1. For local constraints like capacity, we use the mask function in Kool et al. (2019) (see Appendix A.3.4).

**AM as dividing model:** As above mentioned, we could reformulate our dividing model as a sequence-to-sequence model with new reward and global mask function. In this paper, we modify the Attention Model (Kool et al., 2019) as dividing model. The architecture of the dividing model is shown in Figure 4(a). The input and Encoder are the same as AM, while the Decoder is modified to encode our global mask function in Section 4.4. The output of the autoregressive Decoder is the nodes permutation $\boldsymbol{\pi}$, which could divide the VRP into several sub-routes $\boldsymbol{r}$ using Equation 4. The details of AM are shown in Section A.3.3 of Appendix. It should be noted that our main contribution is the two-stage dividing formulation and generalization techniques rather than sequence-to-sequence model. Other models like Graph neural network could also be encoded in TAM.

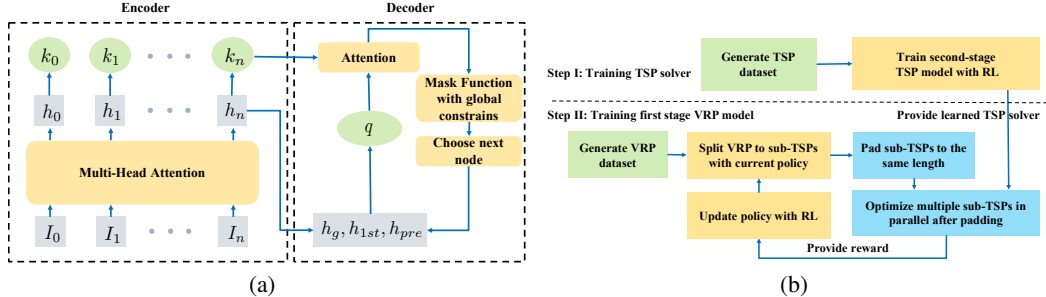

Figure 4: (a) Attention-based model for TAM. (b) Two-step training process of TAM.

## 4.5 TWO-STEP TRAINING ALGORITHM

According to Equation 6, our reward is the sum of the negative optimal length of all sub-routes. However, if we calculate the optimal length of the sub-TSPs with traditional heuristics or exact solvers for training our TAM, two problems arise: 1) we cannot use GPU to accelerate the training process; 2) calculating the optimal solution for each sub-route is relatively time-consuming. To this end, we propose a two-step training algorithm in Figure 4(b) for our TAM by using a learned TSP heuristic to approximately calculate optimal reward. The learned TSP heuristics such as AM have been proved to be effective for small-scale TSPs in previous works (Kool et al., 2019; Bello et al., 2016; Joshi et al., 2020).

**Algorithm** 1 in Appendix A.3.1 reports the implementation details of our training algorithm. The main training process is as follows. **Step I:** we first train a TSP solver with AM using generated dataset for small-scale TSPs. **Step II:** we initialize TAM to split original VRPs in the training dataset into several sub-TSPs. The sub-TSPs are padded to the same length by the proposed padding technique. The sub-TSPs are optimized by the well-trained AM in GPU parallelly after padding. The reward is then calculated using Equation 6 to update the parameters of TAM. The process is repeated iteratively until finding an acceptable policy.

**Padding TSPs to the same length**: The sub-TSPs divided by our TAM have a different number of nodes. Therefore, the length of the input vector is different for different sub-TSPs, which cannot be calculated by GPU in parallel. We borrow the idea padding from CNN (LeCun et al., 2010) to our TAM. We pad the sub-TSPs to the same length by adding depot nodes in sub-TSP input vectors. In this way, we could compute the reward in parallel using the learned TSP model to accelerate the RL training process while with little impact on the optimal reward.

## 4.6 INFERENCE

At inference time, we use the trained TAM to divide the large-scale VRP into several sub-TSPs without padding. Then, the small-scale sub-TSPs could be solved by traditional heuristics in CPU parallelly or by learned heuristics (Algorithm 2 and 3). If we use learned TSP heuristics as second-stage solver, we train several models on datasets of TSP 20, 50, and 100, respectively. At the inference time, we choose the trained TSP model with the closest nodes number to the sub-TSPs as solver. **Search strategy** in the first stage also matters at the inference time. We use three strategies in our TAM: greedy search, sample search, and beam search. The details of the three strategies are described in Appendix A.3.2. In our experiments, we use beam search with beam size 10 as the default strategy.

## 5 EXPERIMENTS

### 5.1 DATA-GENERATION

We set maximum vehicle number $l_m$ as following:

$$l_m = \lceil \frac{\sum_{i \in \Omega} d_i}{C} \rceil + 1 \tag{8}$$

We follow the Nazari et al. (2018); Kool et al. (2019) in the dataset generation for CVRP 100, 400, 1000, 2000, 5000, and 7000. We sample the coordinates of depot and $n$ nodes with uniform

Table 1: Route length and solving time (second) of CVRP with our TAM and benchmark methods that are trained on CVRP100 dataset

| CVRP | METRIC | TAM-AM | TAM-LKH3 | TAM-ORTOOLS | AM | POMO | DPDP | ORTOOLS | LKH3 |
|------|--------|--------|----------|-------------|-----|------|------|---------|------|
| 2000 | LENGTH | 74.31± 1.42 | **64.78**± 1.18 | 65.12±1.18 | 114.36± 2.15 | 485.42±20.12 | - | 68.47± 0.98 | 64.93±1.48 |
|      | TIME   | 2.2±0.00 | 5.63± 0.11 | 11.10±0.69 | 1.87± 0.00 | 4.42±0.08 | - | 100.01± 0.00 | 20.29±0.05 |
| 1000 | LENGTH | 50.06± 0.98 | **46.34**± 0.84 | 46.47±0.84 | 61.42± 1.76 | 160.43±7.21 | - | 48.81± 0.75 | 46.44±0.84 |
|      | TIME   | 0.76± 0.00 | 1.82± 0.03 | 6.03±0.01 | 0.59± 0.00 | 2.27± 0.08 | - | 50.00± 0.00 | 6.15± 0.01 |
| 400  | LENGTH | 27.03±0.51 | 25.93± 0.46 | 25.96±0.47 | 29.33± 0.59 | 29.93±.34 | - | 27.01± 0.47 | 24.67±0.46 |
|      | TIME   | 0.30±0.00 | 1.35± 0.03 | 4.83±0.00 | 0.20± 0.00 | 1.80±0.00 | - | 30.36± 0.16 | 4.10±0.01 |
| **100** | LENGTH | 16.19±0.34 | 16.08±0.33 | 16.08±0.33 | 16.42±0.36 | 15.69±0.26 | 16.57±0.37 | 16.68± 0.36 | 15.58±0.34 |
|      | TIME   | 0.09±0.00 | 0.86± 0.01 | 3.38±0.00 | 0.06± 0.00 | 0.60±0.00 | 0.8±0.01 | 20.03± 0.01 | 2.12±0.00 |

distribution from the domain of [0 1]. The demand of each node is normalized by vehicle capacity. We sample 1280000 CVRP100 instances on the fly as training datasets. And then we test our TAM and benchmark algorithms on 100 instances for CVRP 100, 400, 1000, 2000, 5000, and 7000. We trained second-stage TSP model on datasets of TSP 20, 50, 100. More details are shown in Appendix A.2.1.

## 5.2 CONFIGURATION

For a fair comparison, all our training hyperparameters are the same as AM. We trained TAM for 100 epochs with batch size 512 on the generated dataset. We choose 3 layers transformer in the encoder with 8 heads. The learning rate is constant $\eta = 10^{-4}$. All models are trained on a single GPU Tesla V100. The rest of the parameters such as dimension of hidden state is listed in Kool et al. (2019). We compare our methods (TAM-AM, TAM-LKH3, TAM-Ortools) with three learn-to-construct benchmarks (AM, POMO, DPDP) and two traditional heuristic benchmarks (LKH3 and Ortools).

**AM**: data-driven Attention Model (Kool et al., 2019). **POMO**: Policy Optimization with Multiple Optima, which is an extension of AM with diverse rollouts (Kwon et al., 2020). **DPDP**: Deep Policy Dynamic Programming, which generates VRP solutions from a trained heat-map (Kool et al., 2021).

**LKH3**: An open-source VRP solver based on Lin-Kernighan-Helsgaun algorithm (Helsgaun, 2017; 2009). **Ortools**: An open-source heuristic solver from Google.

**TAM-AM**: Our TAM with AM as the second stage solver. **TAM-LKH3**: Our TAM with LKH3 as the second stage solver. **TAM-Ortools**: Our TAM with Ortools as the second stage solver.

## 5.3 RESULTS ON CVRP

We first report average testing results on 100 instances in Table 1. The test performance on CVRP 400, 1000, and 2000 shows the generalization ability of the method. **For our TAM methods**, we found that TAM-LKH3 obtains the best performance, while TAM-AM generates solution fastest. **Compared with three learn-to-construct methods**, our TAM-LKH3 always generates the best solution in less than 6s and outperforms AM and POMO significantly for CVRP with over 400 nodes, while the DPDP method can not generalize to large-scale VPRs directly. For CVRP 2000, our TAM-LKH3 is over 50% better than AM and POMO. We also observe that the gap between our method and AM and POMO is enlarging as the number of nodes increases. **Compared with traditional heuristics**, we found that our TAM outperforms LKH3 and Ortools in both solution quality and solving time for CVRP 1000 and 2000. Although the LKH3 obtains better solutions for CVRP 400, it takes about 4 × computation time than our TAM. These results demonstrate that: 1) our TAM could learn the sub-routes distribution and generalize the learned heuristic to solve large-scale VRP in seconds; 2) the generalization ability of our TAM outperforms the SOTA learn-to-construct methods such as AM, POMO, and DPDP; 3) our TAM is competitive with traditional heuristics like Ortools and LKH3, while solving time is over 4 × faster. The example of CVRP 2000 in Figure 5(a) of Appendix shows that good solutions usually have more decoupled sub-routes, which is consistent with our experience. These observations indicate our TAM could find potential patterns for good VRP solutions. More examples such as CVRP 1000 are shown in Appendix A.2.6.

**Scalability study:** To validate the scalability performance of our TAM, we further scale our model trained on CVRP100 datasets to CVRP 5000 and 7000. Figure 3 shows the performance of TAM-

Table 2: Route length and solving time (second) of real-world CVRP with our TAM and benchmark methods that are trained on CVRP100 dataset.

| CVRP | METRIC | TAM-AM | TAM-LKH3 | AM | LKH3 |
|---|---|---|---|---|---|
| 1040 | LENGTH | 21.28 | 20.29 | 29.35 | 31.08 |
| | TIME | 3.43 | 5.94 | 3.17 | 50.00 |
| 1299 | LENGTH | 28.85 | 22.06 | 40.56 | 26.69 |
| | TIME | 3.72 | 10.87 | 3.64 | 50.33 |
| 963 | LENGTH | 12.68 | 12.05 | 44.34 | 13.04 |
| | TIME | 3.31 | 6.74 | 2.96 | 50.00 |
| 864 | LENGTH | 24.77 | 23.33 | 42.22 | 39.79 |
| | TIME | 3.05 | 5.61 | 2.66 | 50.00 |
| 680 | LENGTH | 42.32 | 40.02 | 58.05 | 48.45 |
| | TIME | 2.38 | 6.84 | 2.09 | 30.00 |
| 821 | LENGTH | 19.26 | 18.62 | 34.29 | 18.46 |
| | TIME | 2.81 | 5.34 | 2.56 | 100.35 |
| 817 | LENGTH | 15.87 | 15.56 | 26.70 | 14.99 |
| | TIME | 2.74 | 4.69 | 2.49 | 100.22 |
| AVERAGE(926) | **LENGTH** | **23.58** | **21.70** | **39.36** | **27.50** |
| | **TIME** | **3.06** | **6.58** | **2.80** | **61.56** |

LKH3, TAM-AM, AM, and LKH3 against the size of VRP. Both the route length and solution time are averaged on 100 testing instances. Figure 3(a) shows that the gap between our TAM-LKH3 and benchmark methods (AM and LKH3) is enlarging as the size of VRP increases. For CVRP 5000 and 7000, both the TAM-AM and TAM-LKH3 are better than LKH3. In particular, the route length of TAM-LKH3 outperforms LKH3 about 18% on CVRP 5000 while the solving time is about 11% of LKH3. For CVRP 7000, the route length of TAM-LKH3 outperforms LKH3 about 20% while the solving time is about 6.6% of LKH3. Figure 3(b) shows that the solution time of TAM-LKH3, TAM-AM, and AM are stable while the solution time of the traditional heuristics LKH3 increases quickly. These results show our TAM has much better scalability performance than AM and LKH3. The example results for CVRP 7000 and CVRP 5000 are shown in the 7 and Figure 8 respectively.

**Ablation studies** about how our three techniques work are shown in Appendix A.2.5. The comparisons with **Learn-to-improve method and HGS** are shown in Appendix A.2.2 and A.2.3. The results on **CVRPLIB** are shown in Appendix A.2.4. **Training time** is shown in Appendix A.2.7.

## 5.4    RESULTS ON REAL-WORLD CVRPS

For now, we investigate the performance of TAM on synthetic datasets with uniform distribution of customers. However, the distribution of customers in real world is randomly and irregularly clustered. To this end, we report the generalization of our model on real-world CVRPs. **Our TAM has been used for online application.** Table 2 in Appendix summarizes the performance of our TAM, AM, and LKH3 on seven real-world cases. On average, both our TAM-AM and TAM-LKH3 significantly outperform AM and LKH3. The solving time of LKH3 is about 10 times of our TAM-LKH3. Figure 5(b) in Appendix shows the results of a real-world CVRP with 1040 customers. Our TAM-LKH3 finds the best route with length 20.29. The solution time is only about 10% of Ortools and LKH3. Our TAM-AM model finds a good route within 3 seconds, whose length is over 30% shorter than AM. These results mean our model also has a good generalization on real-world large-scale VRPs. The visualization of solutions on the rest real-world VRPs is shown in Appendix A.2.6.

## 6    CONCLUSION

In this paper, we propose generating a sub-routes sequence instead of a nodes sequence to improve the generalization of the learned VRP model. Based on this idea, we propose a two-stage divide model (TAM) to learn how to divide the large-scale VRPs into several small-scale TSPs, which could be solved quickly in parallel. We show that our TAM trained on VRP 100 could generalize to solve large-scale VRPs with over 5000 nodes in real-time. The generalization of our TAM outperforms previous learn-to-construct heuristics on both synthetic and real-world cases. We also encode prior information like maximum vehicle number and expert knowledge like expert-designed heuristics into the data-driven model to improve its generalization. In doing so, we have opened up a new door to generalize the learn-to-construct heuristics and apply them to solve various real-world cases. More discussions about the main limitations and future works could be found in Appendix A.6.

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

# A   APPENDIX

## A.1   VRP FORMULATION

The CVRP (capacitated vehicle routing problem) with maximum vehicle number constraints $l_m$ can be formulated as a mixed integer programming model:

$$\min \sum_{i \in \Omega, j \in \Omega, i \neq j} c_{ij} z_{ij} \tag{9}$$

$$\sum_{j \in \Omega, i \neq j} z_{ij} = 1, \forall i \in \Omega \setminus \{0\} \tag{10}$$

$$\sum_{i \in \Omega, i \neq j} z_{ij} = 1, \forall j \in \Omega \setminus \{0\} \tag{11}$$

$$u_j - u_i \geq d_j - C(1 - z_{ij}), \forall i, j \in \Omega \setminus \{0\}, i \neq j, d_i + d_j \leq C \tag{12}$$

$$0 \leq u_i \leq C - d_i, i \in \Omega \tag{13}$$

$$\sum_{j \in \Omega \setminus \{0\}} z_{j0} = l \tag{14}$$

$$\sum_{i \in \Omega \setminus \{0\}} z_{i0} = l \tag{15}$$

$$l \leq l_m \tag{16}$$

$$z_{ij} \in \{0, 1\}, i \in \Omega, j \in \Omega, i \neq j \tag{17}$$

where equation 9 is objective function, $z_{ij}$ is binary decision variable that has value 1 if the arc from node $i$ to node $j$ is part of the solution and 0 otherwise, $c_{ij}$ is the Euclidean distance from node $i$

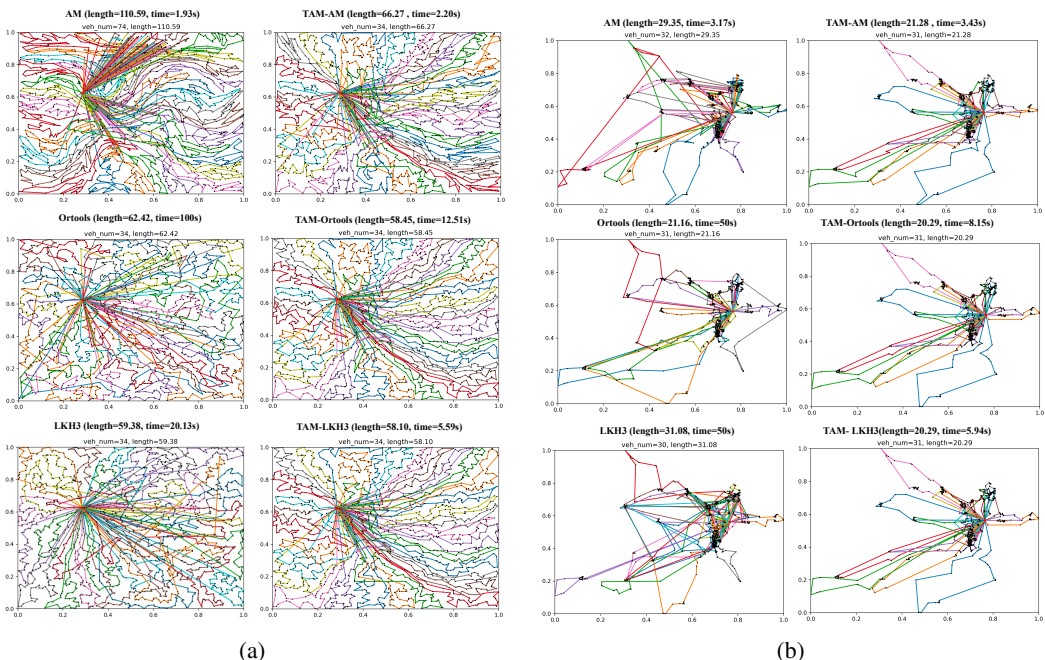

(a)                                                                (b)

Figure 5: The example results of generalizing to the synthetic CVRP 2000 (a) and real-world CVRP 1040 (b) with our TAM and benchmark methods that are trained on CVRP 100 datasets. The examples show that our TAM significantly outperforms the learn-to-construct method AM with similar solving time while outperforming traditional heuristics (LKH3 and Ortools) on both route length and solving time. The methods are as follows: *Top left:* Data-driven Attention Model (**AM**). *Medium left:* **Ortools**. *Bottom left:* **LKH3**. *Top right:* Our TAM with Attention Model as the second-stage solver (**TAM-AM**). *Medium right:* Our TAM with Ortools as the second-stage solver (**TAM-Ortools**). *Bottom right:* Our TAM with LKH3 as the second-stage solver (**TAM-LKH3**).

Table 3: Route length and solving time (second) of CVRP using TAM-LKH3 and Learn-to-Improve (L2I) that are trained on CVRP100 dataset.

| CVRP | METRIC | L2I | TAM-LKH3 |
|---|---|---|---|
| 2000 | LENGTH | 138.75 | 64.78 |
| | TIME | 25.38 | 5.63 |
| 1000 | LENGTH | 93.15 | 46.34 |
| | TIME | 6.32 | 1.82 |
| 400 | LENGTH | 30.67 | 25.93 |
| | TIME | 3.12 | 1.35 |
| **100** | LENGTH | 16.62 | 16.08 |
| | TIME | 1.01 | 0.86 |

to node $j$; equation 10 and 11 ensure every customer is visited once; equation 11 and 12 define the capacity and connectivity constraints, $u_i$ represents the demand in the vehicle after visiting customer $i$; equation 13, 14, and 15 ensure the maximum vehicle number constraints.

## A.2 MORE RESULTS

### A.2.1 MORE DETAILS OF EXPERIMENT CONFIGURATION

The capacity for synthetic instances of CVRP 100, 400, 1000, 2000, 5000, and 7000 is set to 50, 150, 200, 300, 300, and 300, respectively. At the testing stage, a random seed with a value of 1234 is utilized to generate the instances. At the training stage, we use the Attention Model trained on TSP 20 dataset as default second stage solver. The default search strategy is beam search with beam size 2.

### A.2.2 COMPARISON WITH LEARN-TO-SEARCH METHOD

As mentioned in related work, the learn-to-search methods could obtain competitive results compared with traditional heuristics such as LKH3 when given enough searching time. In this section, we compare the generalization ability and solving time of our method TAM-LKH3 and a well-known learn-to-improve method (L2I, 2019) (Lu et al., 2019). The results are shown in Table 3. We train the L2I model on the dataset of CVRP100 with the same parameters as Lu et al. (2019) and then generalize the L2I model to solve CVRP 400, 1000, and 2000 in real-time and zero-shot settings. For CVRP 100, we found that our TAM-LKH3 outperforms L2I in real-time. We observe that the solution quality of L2I degrades from 15.57 (Lu et al., 2019) to 16.62 when searching in time. For larger VRP instances, the gap of solution quality between our TAM-LKH3 and L2I enlarges significantly, such as 138.75 (L2I) versus 64.78 (TAM-LKH3) for CVRP 2000. These results suggest that: 1) our TAM-LKH3 outperforms L2I on both generalization ability and solving time; 2) the performance of L2I will degrade a lot in real-time and zero-shot generalization settings.

In addition, the searching speed of learn-to-search methods and traditional heuristics heavily depends on the quality of initial solution, while our method can generalize to generate large-scale VRP solutions in real-time. Therefore, we could search better solutions for large-scale VRPs by taking advantages of both our TAM and learn-to-search method.

### A.2.3 COMPARISON WITH HGS ON LARGE-SCALE VRPS

We have tested our TAM and the open-sourced HGS on 100 instances of CVRP 5000 and 7000 in real-time. Table 6 and 7 report the average route length and solving time of the instances. HGS-t60 means the HGS with 60s time-limit. We also implement the TAM-HGS, which uses HGS as the second stage solver of TAM. For real-time application, we found the HGS cannot find a solution in 60 seconds while TAM-HGS could find a good solution in 30.23s and 52.36s for CVRP 5000 and CVRP 7000 respectively. It's also interesting to see that TAM-HGS consumes more time than TAM-LKH3 while finding the nearly same quality solutions.

Table 4: Route length and solving time (second) of CVRP using our TAM-Ortools without new reward or without maximum vehicle number mask function that are trained on CVRP100 dataset.

| CVRP | METRIC | TAM-ORTOOLS | W/O NEW REWARD | W/O MASK FUNCTION |
|---|---|---|---|---|
| 2000 | LENGTH | $65.12 \pm 1.179$ | $66.27 \pm 1.222$ | $69.52 \pm 1.270$ |
| | TIME | $11.10 \pm 0.694$ | $11.32 \pm 0.601$ | $11.32 \pm 0.572$ |
| 1000 | LENGTH | $46.47 \pm 0.843$ | $47.23 \pm 0.881$ | $48.24 \pm 0.897$ |
| | TIME | $6.03 \pm 0.005$ | $6.15 \pm 0.006$ | $6.13 \pm 0.006$ |
| 400 | LENGTH | $25.96 \pm 0.465$ | $26.10 \pm 0.465$ | $26.03 \pm 0.457$ |
| | TIME | $4.83 \pm 0.003$ | $4.73 \pm 0.009$ | $4.76 \pm 0.002$ |
| **100** | LENGTH | $16.08 \pm 0.330$ | $16.09 \pm 0.341$ | $16.06 \pm 0.340$ |
| | TIME | $3.38 \pm 0.002$ | $3.35 \pm 0.002$ | $3.34 \pm 0.003$ |

Table 5: Route length and solving time (second) of CVRP using TAM (without a second stage solver) and AM that are trained on CVRP100 dataset.

| CVRP | METRIC | AM | TAM |
|---|---|---|---|
| 2000 | LENGTH | $114.36 \pm 2.148$ | $\mathbf{87.14 \pm 1.489}$ |
| | TIME | $1.87 \pm 0.003$ | $1.85 \pm 0.002$ |
| 1000 | LENGTH | $61.42 \pm 1.757$ | $\mathbf{57.76 \pm 1.070}$ |
| | TIME | $0.59 \pm 0.002$ | $0.58 \pm 0.001$ |
| 400 | LENGTH | $\mathbf{29.33 \pm 0.589}$ | $30.77 \pm 0.580$ |
| | TIME | $0.20 \pm 0.002$ | $0.19 \pm 0.001$ |
| **100** | LENGTH | $\mathbf{16.42 \pm 0.360}$ | $16.71 \pm 0.354$ |
| | TIME | $0.06 \pm 0.002$ | $0.05 \pm 0.000$ |

### A.2.4 RESULTS ON LARGE-SCALE INSTANCES OF CVRPLIB

We also test our TAM on all large-scale instances with 500-20000 nodes on Uchoa et al. (2014) and Arnold, Gendreau and Sörensen (2017) of CVRPLIB Uchoa et al. (2017). Table 8 and 9 report the gap and solving time of the instances. The results show that our TAM-LKH3 could find significantly better results than AM and could find a better solution much faster than LKH3. We also observed that the gap between our TAM and the best solution is about 5-10% when generalizing to VRP with about 1000 nodes, and about 15-25% when gneralizing to VRP with over 3000-10000 nodes. This is acceptable for real-time application given that we limit the maximum solving time of our TAM as 60 seconds for instances with less than 10000 nodes and the beam search size as 10 for real-time application. There are two main reasons behind the gap difference: 1) larger-scale VRP is much harder to solve; 2) the difference between large-scale VRP and VRP 100 (used for training TAM) enlarges as the nodes number increases. If the optimal solutions are required for non-real-time applications, we should significantly improve the sample size of our TAM to find better solution.

### A.2.5 ABLATION STUDY

To study how the proposed reward and global mask function affect the performance of our TAM, Table 4 reports our testing results using three methods, the original TAM-Ortools, the TAM-Ortools trained with standard reward, and the TAM-Ortools trained without the global mask.

**1) Our reward vs distance reward**: We found that for CVRP 100, the performance gap between TAM-Ortools with and without new reward are small because we both train and test on CVRP 100 dataset. As we generalize the learned model to CVRP 400, 1000, and 2000, the gaps are increasing to 0.5%, 1.6%, and 1.8%, respectively. The results mean our new reward could obviously improve the generalization ability of our TAM to large-scale VRPs and has little impact on its performance on small-scale VRPs.

**2) With vs without global mask function**: The possible minimal vehicle number for CVRP are $l_m - 1$(in Equation 8). Therefore, the $l_m$ provides prior information for minimal vehicle number.

Table 6: Route length of large-scale synthetic CVRPs using TAM and HGS.

| LENGTH | AM | TAM-AM | LKH3 | TAM-LKH3 | TAM-HGS | HGS-T60 |
|---|---|---|---|---|---|---|
| CVRP-5000 | 257.06 | 172.22 | 175.66 | 144.64 | 142.83 | - |
| CVRP-7000 | 354.28 | 233.44 | 244.97 | 196.91 | 193.64 | - |

Table 7: Solving time of large-scale synthetic CVRPs using TAM and HGS.

| TIME(S) | AM | TAM-AM | LKH3 | TAM-LKH3 | TAM-HGS | HGS-T60 |
|---|---|---|---|---|---|---|
| CVRP-5000 | 11.50 | 11.78 | 151.64 | 17.19 | 30.23 | 60.00 |
| CVRP-7000 | 26.13 | 26.47 | 501.26 | 33.21 | 52.36 | 60.00 |

For the results in Table 4, we observe that adding our global mask function could improve results of TAM-Ortools in CVRP 400, 1000, 2000 about 0.3%, 3.8%, 6.8%, respectively. The gaps are increasing as the number of nodes increases. This is because the generalization-ability of the learned heuristics without the global mask is decreasing as the number of nodes increases. Thus, the number of vehicles has a larger impact on the total route length for larger VRPs. These results indicate our global mask function could significantly improve the generalization ability of the learned model.

**3) TAM vs AM**: Table 5 reports the results of TAM without a second stage solver and AM on CVRP 100, 400, 1000, and 2000. We found that for smaller-scale CVRP 100, and 400, the AM outperforms TAM slightly. This is because the dividing model does not care about the order of nodes inside the sub-route. However, for large-scale VRPs such as CVRP 1000 and 2000, our TAM still outperforms AM significantly even without a second-stage solver (23.8% for CVRP 200). These results indicate the superior generalization ability of our TAM.

### A.2.6 MORE CVRP INSTANCES

We report more synthetic and real-world CVRP instances for comparison and understanding of our method. Figure 7, 8, 9 show the results on synthetic CVRP 7000, 5000, 1000 respectively. Figure 10, 11, 12, 13, 14, 15 show the results on some real-world and large-scale CVRP instances, respectively. Table 2 summrizes the performance of our TAM, AM, and LKH3 on seven real-world cases. The datasets of location, demand, and vehicle capacity are all from real-world cases. For anonymity, we normalized the location and demand using the same method as the synthetic dataset. We found that our TAM-LKH3 could always find good solutions in real-time, while LKH3 fails to search for good solutions without enough searching time in some real-world cases (Figure 10, 11, 12, 13). If given enough time for iterations (not real-time), traditional heuristics like LKH3 could find good solutions (Figure 14, 15). We also found that good solutions for large-scale VRPs are always visually attractive and decoupled for each sub-routes, while bad solutions usually have lots of crossovers among sub-routes. These results indicate the robustness of our TAM on synthetic and real-world cases.

### A.2.7 TRAINING TIME COMPARISON BETWEEN TAM AND AM

Because our TAM uses simliar hyperparameters as AM, the only difference that matters for training time is reward calculation. The AM reward is calculated using the distance function directly, while our reward is calculated by a learned TSP solver in GPU. Thus, our training time is a bit larger than AM. The training time of our TAM and AM for 100 epochs are shown in Table 10.

Table 8: Gap of CVRPLIB instances

| Length | No. Nodes | AM | TAM-AM | LKH3 | TAM-LKH3 |
|---|---|---|---|---|---|
| X-n502-k39 | 502 | 13.97% | 6.45% | 1.02% | 5.75% |
| X-n513-k21 | 513 | 18.13% | 11.47% | 3.50% | 9.41% |
| X-n524-k153 | 524 | 27.57% | 16.73% | 3.91% | 14.34% |
| X-n536-k96 | 536 | 27.67% | 10.32% | 19.05% | 10.25% |
| X-n548-k50 | 548 | 28.09% | 14.30% | 5.38% | 13.71% |
| X-n561-k42 | 561 | 17.85% | 10.60% | 8.33% | 9.36% |
| X-n573-k30 | 573 | 25.79% | 10.50% | 4.77% | 7.56% |
| X-n586-k159 | 586 | 19.40% | 8.90% | 12.73% | 8.89% |
| X-n599-k92 | 599 | 24.02% | 9.56% | 34.68% | 9.37% |
| X-n613-k62 | 613 | 18.20% | 11.64% | 16.38% | 9.69% |
| X-n627-k43 | 627 | 51.64% | 10.31% | 17.23% | 9.61% |
| X-n641-k35 | 641 | 24.62% | 8.56% | 7.93% | 7.39% |
| X-n655-k131 | 655 | 20.55% | 7.31% | 3.03% | 6.66% |
| X-n670-k130 | 670 | 31.18% | 19.29% | 18.55% | 19.01% |
| X-n685-k75 | 685 | 19.80% | 11.03% | 17.59% | 10.40% |
| X-n701-k44 | 701 | 29.56% | 7.73% | 8.87% | 6.81% |
| X-n716-k35 | 716 | 32.90% | 12.00% | 9.35% | 10.05% |
| X-n733-k159 | 733 | 18.74% | 8.60% | 16.65% | 8.51% |
| X-n749-k98 | 749 | 17.06% | 10.22% | 19.90% | 10.03% |
| X-n766-k71 | 766 | 29.50% | 9.84% | 12.44% | 8.82% |
| X-n783-k48 | 783 | 15.60% | 12.95% | 11.87% | 10.23% |
| X-n801-k40 | 801 | 34.30% | 11.91% | 5.57% | 10.34% |
| X-n819-k171 | 819 | 16.02% | 9.70% | 28.92% | 9.60% |
| X-n837-k142 | 837 | 14.50% | 7.58% | 11.16% | 7.51% |
| X-n856-k95 | 856 | 12.27% | 8.99% | 8.09% | 8.68% |
| X-n876-k59 | 876 | 33.15% | 9.37% | 9.46% | 8.10% |
| X-n957-k87 | 957 | 36.06% | 16.94% | 10.43% | 16.53% |
| X-n895-k37 | 895 | 36.84% | 14.32% | 29.72% | 9.39% |
| X-n916-k207 | 916 | 17.19% | 8.91% | 10.34% | 8.87% |
| X-n936-k151 | 936 | 27.86% | 13.92% | 22.89% | 13.47% |
| X-n979-k58 | 979 | 30.41% | 8.23% | 15.80% | 6.70% |
| X-n1001-k43 | 1001 | 18.36% | 12.99% | 13.79% | 10.64% |
| Leuven1-3001 | 3001 | 46.93% | 20.24% | 18.10% | 19.30% |
| Leuven2-4001 | 4001 | 53.31% | 38.57% | 22.14% | 15.88% |
| Antwerp1-6001 | 6001 | 39.30% | 24.90% | 24.20% | 24.01% |
| Antwerp2-7001 | 7001 | 50.32% | 33.20% | 31.09% | 22.55% |
| **Average** | **1199** | **27.18%** | **13.00%** | **14.30%** | **11.04%** |
| Ghent1-10001 | 10001 | 46.89% | 30.20% | - | 29.53% |
| Ghent2-11001 | 11001 | 52.20% | 33.29% | - | 23.65% |
| Brussels1-15001 | 15001 | 52.37% | 43.42% | - | 27.15% |
| Brussels2-16001 | 16001 | 52.44% | 39.04% | - | 37.07% |

Table 9: Solving time of CVRPLIB instances.

| TIME (S) | NO. NODES | AM | TAM-AM | LKH3 | TAM-LKH3 |
|---|---|---|---|---|---|
| X-N502-K39 | 502 | 0.74 | 1.01 | 2.05 | 1.47 |
| X-N513-K21 | 513 | 0.68 | 0.99 | 2.10 | 1.41 |
| X-N524-K153 | 524 | 0.88 | 1.00 | 2.78 | 1.30 |
| X-N536-K96 | 536 | 0.83 | 0.95 | 2.37 | 1.28 |
| X-N548-K50 | 548 | 0.78 | 0.88 | 2.05 | 1.18 |
| X-N561-K42 | 561 | 0.80 | 0.93 | 2.08 | 1.31 |
| X-N573-K30 | 573 | 0.79 | 0.99 | 2.05 | 1.97 |
| X-N586-K159 | 586 | 1.00 | 1.08 | 3.59 | 1.58 |
| X-N599-K92 | 599 | 1.00 | 1.13 | 2.33 | 1.52 |
| X-N613-K62 | 613 | 0.92 | 1.02 | 2.34 | 1.39 |
| X-N627-K43 | 627 | 0.98 | 1.01 | 2.15 | 1.71 |
| X-N641-K35 | 641 | 1.03 | 1.53 | 2.10 | 2.03 |
| X-N655-K131 | 655 | 1.07 | 1.13 | 3.63 | 1.63 |
| X-N670-K130 | 670 | 1.09 | 1.24 | 4.97 | 1.64 |
| X-N685-K75 | 685 | 1.01 | 1.16 | 2.62 | 1.72 |
| X-N701-K44 | 701 | 1.03 | 1.16 | 3.04 | 1.51 |
| X-N716-K35 | 716 | 1.05 | 1.23 | 3.06 | 1.92 |
| X-N733-K159 | 733 | 1.25 | 1.36 | 4.82 | 1.89 |
| X-N749-K98 | 749 | 1.61 | 1.55 | 3.42 | 1.95 |
| X-N766-K71 | 766 | 1.16 | 1.34 | 4.28 | 2.01 |
| X-N783-K48 | 783 | 1.14 | 1.35 | 4.06 | 2.74 |
| X-N801-K40 | 801 | 1.46 | 1.63 | 4.12 | 2.47 |
| X-N819-K171 | 819 | 1.50 | 1.81 | 4.65 | 2.44 |
| X-N837-K142 | 837 | 1.37 | 1.49 | 5.34 | 2.06 |
| X-N856-K95 | 856 | 1.30 | 1.40 | 5.27 | 2.10 |
| X-N876-K59 | 876 | 1.29 | 1.43 | 5.20 | 2.66 |
| X-N957-K87 | 957 | 3.42 | 2.91 | 4.21 | 5.01 |
| X-N895-K37 | 895 | 2.64 | 2.58 | 4.23 | 3.91 |
| X-N916-K207 | 916 | 3.16 | 3.09 | 8.93 | 6.84 |
| X-N936-K151 | 936 | 3.94 | 3.41 | 9.22 | 6.05 |
| X-N979-K58 | 979 | 3.03 | 2.95 | 4.22 | 5.06 |
| X-N1001-K43 | 1001 | 2.95 | 2.93 | 4.18 | 4.68 |
| LEUVEN1-3001 | 3001 | 9.73 | 9.82 | 68.99 | 15.59 |
| LEUVEN2-4001 | 4001 | 13.45 | 13.68 | 73.96 | 23.73 |
| ANTWERP1-6001 | 6001 | 12.81 | 12.98 | 596.21 | 24.91 |
| ANTWERP2-7001 | 7001 | 14.96 | 15.26 | 479.47 | 31.90 |
| **AVERAGE** | **1199** | **2.72** | **2.82** | **37.22** | **4.85** |
| GHENT1-10001 | 10001 | 21.41 | 21.66 | - | 37.26 |
| GHENT2-11001 | 11001 | 38.55 | 37.56 | - | 55.73 |
| BRUSSELS1-15001 | 15001 | 131.39 | 139.09 | - | 166.81 |
| BRUSSELS2-16001 | 16001 | 165.94 | 158.71 | - | 187.48 |

## A.3   DETAILS OF METHOD

### A.3.1   TRAINING WITH REINFORCEMENT LEARNING

We have defined our stochastic policy $p_\theta(\boldsymbol{\pi}|s)$ in Equation 4 using an attention-based model in Figure 4(a), from which we could determine a sub-routes sequence and the number of vehicle by

Table 10: Training time comparison between TAM and AM on CVRP100 dataset.

| METHOD | TRAINING TIME (H) | EPOCH NUMBER |
|---|---|---|
| AM | 35.7 | 100 |
| TAM | 78.3 | 100 |

---

**Algorithm 1** Training Algorithm for TAM
___

**Input:** number of epochs $E$, batch size $B$, steps $T$, trained TSP model $AM$
Initialize $\boldsymbol{\theta} = \boldsymbol{\theta}_0, \boldsymbol{\theta}^{BL} = \boldsymbol{\theta}_0$.
**for** epoch $= 1$ **to** $E$ **do**
   **for** step $= 1$ **to** $T$ **do**
      Random generate instances $s_i, \forall i \in \{1, \ldots, B\}$
      Sample split $\boldsymbol{\pi}_i$ using policy $p_{\boldsymbol{\theta}}, \forall i \in \{1, \ldots, B\}$
      Greedily sample split $\boldsymbol{\pi}_i^{BL}$ using policy $p_{\boldsymbol{\theta}^{BL}}, \forall i \in \{1, \ldots, B\}$
      Padding all the sub-TSPs of $\boldsymbol{\pi}_i$ and $\boldsymbol{\pi}_i^{BL}$ to the same length, respectively, $\forall i \in \{1, \ldots, B\}$
      Calculate the loss $L(\boldsymbol{\pi}_i), L(\boldsymbol{\pi}_i^{BL})$ using $AM$ according to Equation 18 in parallel, respectively, $\forall i \in \{1, \ldots, B\}$
      $\nabla\mathcal{L} \leftarrow \sum_{i=1}^{B} \left( L\left(\boldsymbol{\pi}_i\right) - L(\boldsymbol{\pi}_i^{BL}) \right) \nabla_{\boldsymbol{\theta}} \log p_{\boldsymbol{\theta}}\left(\boldsymbol{\pi}_i\right)$
      $\boldsymbol{\theta} \leftarrow \text{Adam}(\boldsymbol{\theta}, \nabla\mathcal{L})$
   **end for**
   **if** $p_{\boldsymbol{\theta}}$ is better than $p_{\boldsymbol{\theta}^{BL}}$ **then**
      $\boldsymbol{\theta}^{\text{BL}} \leftarrow \boldsymbol{\theta}$
   **end if**
**end for**

---

identifying the locations and number of depots of the sampled sequence $\boldsymbol{\pi}$. To train the model, we define a new loss $\mathcal{L}(\boldsymbol{\theta}|s) = \mathbb{E}_{p_{\boldsymbol{\theta}}(\boldsymbol{\pi}|s)}[L(\boldsymbol{\pi})]$. According to Equation 5 and 6, the cost $L(\boldsymbol{\pi})$ is calculated as:

$$L(\boldsymbol{\pi}) = -R = \sum_{i=1}^{l} \min_{\phi} \text{dist}(\phi(r_i)) \tag{18}$$

Then, we could optimize the $\mathcal{L}$ using REINFORCE algorithm to estimate policy gradient:

$$\nabla\mathcal{L}(\boldsymbol{\theta}|s) = \mathbb{E}_{p_{\boldsymbol{\theta}}(\boldsymbol{\pi}|s)}\left[(L(\boldsymbol{\pi}) - b(s))\nabla \log p_{\boldsymbol{\theta}}(\boldsymbol{\pi}|s)\right] \tag{19}$$

where $b(s)$ is greedy rollout baseline to estimate the value function (Kool et al., 2019). If the reward of current split $\boldsymbol{\pi}$ is larger than the baseline, we should push the parameters $\boldsymbol{\theta}$ of policy network to the direction that is more likely to generate the current split $\boldsymbol{\pi}$.

The final training algorithm for TAM is shown in Algorithm 1.

### A.3.2 SEARCH STRATEGY

The search strategy in the first stage also matters at the inference time. We use three strategies in our TAM:

**Greedy Search**: In this strategy, we use the most likely split according to our model, and then optimize all sub-routes in parallel. This strategy could generate acceptable solutions with minimum solving time, which could be used in real-time.

**Beam Search**: In this strategy, we sample $k$, called beam size, most likely splits according to our model, and optimize each sub-route for all $k$ splits. Then we choose the final VRP route with minimum length. With a larger beam size, this strategy could generate a better split, but also consume more time. We found that increasing the beam size could improve our results obviously first, but will saturate because beam search tends to sample similar splits. Besides, the order of nodes inside each sub-route of current split is invariant for the final solution. Therefore, we use 10 as the beam size.

---

**Algorithm 2** Inference algorithm for TAM with traditional heuristics as second-stage solver

---

**Input:** Learned dividing policy $p_\theta$, VRP instance $s$

Divide instance $s$ into several small scale TSPs $\{s_i^{tsp}|i = 1 \ldots l\}$ using policy $p_\theta$ with default search strategy

Use a traditional solver such as LKH-3 to solve the small-scale TSPs parallelly in CPU

Combine the output of the TSPs as the solution of VRP instance $s$

---

**Algorithm 3** Inference algorithm for TAM with learned heuristics as second-stage solver

---

**Input:** Learned dividing policy $p_\theta$, VRP instance $s$, learned TSP solver $AM_{20}$, $AM_{50}$, and $AM_{100}$ on dataset of TSP 20, TSP 50, and TSP 100, respectively

Divide instance $s$ into several small scale TSPs $\{s_i^{tsp}|i = 1 \ldots l\}$ using policy $p_\theta$ with default search strategy

Choose the second-stage solver $AM_j, j \in \{20, 30, 50\}$ for $s_i^{tsp}$ by minimizing the absolute value between the node number of $s_i^{tsp}$ and $j$.

Use the chosen solver to solve the small-scale TSPs parallelly in CPU or GPU

Combine the output of the TSPs as the solution of VRP instance $s$

---

**Sample Search**: In this strategy, we sample $k$, called sample size, splits randomly according to our model, and optimize each sub-route for all $k$ splits. Then we choose the final VRP route with minimum length. This strategy could sample totally different splits and could greatly improve the results of our TAM with a large sample number. However, the computation time and memory are also increasing with the sample number. For instance, Figure 16 shows that TAM-LKH3 with sample size 1000 could find the best solution 6.84 among all methods for a real-world CVRP 388 instance, while TAM-LKH3 with beam size 10 find a solution 6.99. However, the solving time also increases from 3.73s to 17.52s.

We report the results of the greedy search of our TAM in Table 11. Comparing Table 1 and Table 11, we found that beam search with beam size 10 outperforms greedy search about 2.6% for our TAM-Ortools, about 3.3% for our TAM-AM on CVRP 2000. The gap between the beam search and greedy search is larger as the number of nodes increases. Besides, the solving time of beam search is close to that of greedy search. These results mean beam search could effectively improve the generalization of our TAM on solving large-scale VRPs in real-time. It should be noted that the beam search could also significantly improve the performance of AM. For CVRP 2000, the route length of AM reduces from 178.31 to 114.36 after using beam search.

### A.3.3 ATTENTION MODEL

To learn the policy in Equation 1, AM uses an attention-based Encoder-Decoder model to approximate the policy. The Encoder is a transformer model without positional encoding which transforms each

Table 11: Route length and solving time (second) of CVRP using TAM (greedy search), AM (greedy search), and Ortools.

| CVRP | METRIC | AM | TAM-AM | TAM-ORTOOLS | ORTOOLS |
|---|---|---|---|---|---|
| 2000 | LENGTH | $178.31 \pm 27.601$ | $76.87 \pm 2.293$ | $66.87 \pm 2.096$ | $68.47 \pm 0.984$ |
|      | TIME | $0.57 \pm 0.005$ | $0.82 \pm 0.006$ | $10.86 \pm 0.850$ | $100.01 \pm 0.001$ |
| 1000 | LENGTH | $65.44 \pm 5.232$ | $50.38 \pm 1.003$ | $46.49 \pm 0.842$ | $48.81 \pm 0.749$ |
|      | TIME | $0.26 \pm 0.002$ | $0.44 \pm 0.002$ | $5.71 \pm 0.004$ | $50.00 \pm 0.001$ |
| 400 | LENGTH | $29.38 \pm 0.504$ | $27.07 \pm 0.517$ | $25.97 \pm 0.473$ | $27.01 \pm 0.468$ |
|     | TIME | $0.10 \pm 0.002$ | $0.27 \pm 0.001$ | $4.67 \pm 0.002$ | $30.36 \pm 0.157$ |
| 100 | LENGTH | $16.71 \pm 0.383$ | $16.24 \pm 0.335$ | $16.12 \pm 0.329$ | $16.68 \pm 0.358$ |
|     | TIME | $0.03 \pm 0.002$ | $0.15 \pm 0.002$ | $3.17 \pm 0.002$ | $20.03 \pm 0.014$ |

node input $I_i$ to hidden embedding $\mathbf{h}_i$. The $\mathbf{h}_g$ is the global embedding of VRP, by averaging on the hidden embeddings of all nodes. Then, the Decoder uses $\mathbf{h}_c$ as context embedding to query hidden states of unvisited nodes to choose the next node to serve (Kool et al., 2019):

$$\mathbf{h}_c = [\mathbf{h}_g, \mathbf{h}_{1st}, \mathbf{h}_{pre}] \tag{20}$$

where $\mathbf{h}_{1st}$ represents the hidden state of first visited node; $\mathbf{h}_{pre}$ is the hidden state of last visited node. The query, key, and value of $i^{th}$ node for the Decoder are as follows:

$$\mathbf{q} = W^Q \mathbf{h}_c \quad \mathbf{k}_i = W^K \mathbf{h}_i, \quad \mathbf{v}_i = W^V \mathbf{h}_i \tag{21}$$

Then, we could compute the possibility to visit node $i$ by attention as following:

$$p_i = p_\theta\left(\pi_t = I_i | s_t\right) = \text{softmax}(u_i) \tag{22}$$

$$u_i = \begin{cases} \tanh\left(\frac{\mathbf{q}^T \mathbf{k}_i}{\sqrt{dim_k}}\right) & \text{if node } i \text{ is unvisited or depot} \\ -\infty & \text{otherwise.} \end{cases} \tag{23}$$

where $dim_k$ is the dimension of hidden state.

More details about AM could be found in Kool et al. (2019).

### A.3.4  LOCAL MASK FUNCTIONS

For local constraints like capacity, we could use the same mask function in Kool et al. (2019) and Bello et al. (2016). If the rest capacity of the current vehicle $(C - c)$ cannot serve the demand of node $i$ $(d_i)$, then the current vehicle cannot visit node $i$:

$$u_i = \begin{cases} \tanh\left(\frac{\mathbf{q}^T \mathbf{k}_i}{\sqrt{dim_k}}\right) & \text{if } c + d_i \leq C \\ -\infty & \text{otherwise.} \end{cases} \tag{24}$$

where $c$ is the used capacity of the current vehicle.

For time window constraints, it could be considered in TAM with the following mask function. If the current vehicle cannot arrive the node $i$ in the time window $[T_i^l, T_i^u]$, then the current vehicle cannot visit node $i$:

$$u_i = \begin{cases} \tanh\left(\frac{\mathbf{q}^T \mathbf{k}_i}{\sqrt{dim_k}}\right) & \text{if } T_i^l \leq T + T^s + t_i \leq T_i^u \\ -\infty & \text{otherwise.} \end{cases} \tag{25}$$

where $T$ is arrival time of current nodes, $T^s$ is service time of current nodes, $t_i$ is the travel time from current nodes to the node $i$.

### A.4  THEORETICAL ANALYSIS

### A.4.1  PROOF FOR GLOBAL MASK FUNCTION

**Theorem 1** (Global mask function for maximum vehicle number constraint). *Assume that 1) the total capacity of vehicles is larger than the total demand of customers $l_m * C > \sum_i d_i$; 2) the maximum demand of a single customer is much less than the capacity of vehicle $\max_{i \in \{1,...,n\}} d_i \ll C$. Then, the proposed global mask function enforces the satisfaction of the maximum vehicle number constraint $l \leq l_m$.*

**Proof.** Note that our global mask enforces the current vehicle cannot return to the depot (cannot use new vehicles) if the capacity of unused vehicles cannot serve the rest demand. In the extreme scenario, our mask function prevents the current vehicle to return to the depot unless it visits as many customers as possible.

1) If the total demand $\sum_i d_i$ is less than the current vehicle's capacity $C$, our global mask function enforces using the current vehicle to serve all customers.

2) If the total demand $\sum_i d_i$ is larger than the current vehicle's capacity $C$, there always exists a subset of customers $\Omega_s$ that are chosen by our dividing model and a customer $k$ to satisfy the following inequality:

$$\sum_{i \in \Omega_s} d_i \leq C \leq \sum_{i \in \Omega_s} d_i + d_k \tag{26}$$

Dividing $C$ from above inequality:

$$\sum_{i \in \Omega_s} d_i/C \le 1 \le \sum_{i \in \Omega_s} d_i/C + d_k/C \tag{27}$$

According to Assumption 2, the following inequality holds:

$$d_k/C \le \max_{i \in \{1,\ldots,n\}} d_i/C \approx 0 \tag{28}$$

Therefore, there exists a subset of customers $\Omega_s$ that can be exactly served by one vehicle, which satisfies the following Equation:

$$C = \sum_{i \in \Omega_s} d_i \tag{29}$$

Minus above Equation 29 from Assumption 1, Thus,

$$(l_m - 1) * C > \sum_i d_i - \sum_{i \in \Omega_s} d_i = \sum_{i \in \Omega_u} d_i \tag{30}$$

$\Omega_u$ is the set of unvisited customers. The left side is the remaining capacity. The right side is the remaining demand. Therefore, the rest capacity of vehicles could still serve the rest demand, which satisfies our proposed global mask function. Therefore, the current vehicle could return to the depot and use the next vehicle. Repeat the above process, we will always obtain a VRP solution while satisfying the maximum vehicle number constraints.

*Remark:* In terms of Assumption 1, the total capacity of available vehicles is usually large than the total demand of customers for the real-world VRPs with redundancy, because we want to fulfill all the orders and keep the promise to the customers. That's one of the reasons why we set the maximal vehicle capacity as Equation 8. In terms of Assumption 2, the single demand is usually much less than the capacity of vehicles in the real-world considering a large number of customers. It should be noted that in some rare scenarios, the second Assumption 2 does not hold. To handle this situation, we will let the current vehicle visit as many customers as possible before returning to the depot. In the early training state, we will add a new vehicle when encountering the situation. In the testing state, given the well-trained policy, search strategy, and the redundant vehicle capacity in Equation 8, all solutions of our testing cases are feasible.

### A.4.2 ANALYSIS FOR GENERALIZATION PERFORMANCE

The idea behind the sequence-to-sequence model is learning a stochastic policy $p(\pi_t|s_t)$, that could determine the next customer $\pi_t$ to visit given current state $s_t$. Then, the optimal route $\boldsymbol{\pi}$ could be modeled as:

$$p(\boldsymbol{\pi}|s) = \prod_{t=1}^{n+l} p\left(\pi_t|s_t\right) \tag{31}$$

However, our TAM aims to find a stochastic policy determining the next route $r_t$ to visit and then use the traditional and learned heuristics to optimize the route. The stochastic policy for optimal split $\boldsymbol{r}$ is modeled as:

$$p_\theta(\boldsymbol{r}|s) = \prod_{t=1}^{l} p_\theta\left(r_t|s_t\right) \tag{32}$$

To compare the two policies, we reformulated the Equation 31 as:

$$p(\boldsymbol{\pi}|s) = \left(\prod_{t=1}^{l} p_\theta\left(r_t|s_t\right)\right)\left(\prod_{t=1}^{l}\prod_{i=1}^{l_t} p_\beta\left(\pi_i|r_t, \pi_{t,0:i-1}\right)\right) \tag{33}$$

This formulation means we first obtain optimal sub-route split and then determine the order of customers in the sub-route. The $l_t$ is number of node in the route $r_t$. The $\beta$ denotes the parameters of

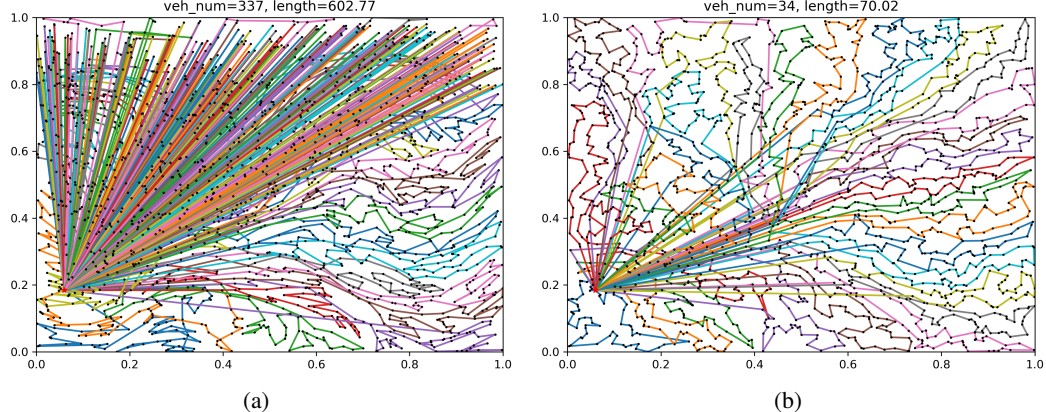

Figure 6: A generalization case for AM and TAM-LKH3 on the same CVRP 2000 instance with model trained on CVRP 100 datasets. The vehicle number of AM is about 10 times larger than that of TAM. (a) The generalization result using AM. (b) The generalization result using TAM-LKH3.

the policy that determines the customer's order inside a sub-route. The $\pi_{t,0:i-1}$ represents the chosen partial sequence inside sub-route $r_t$.

With the above formulation, we argue that all three proposed techniques matter for the generalization of TAM.

**1) Generating sub-route sequence and optimizing in parallel**

Comparing Equation 33 and Equation 32, we found that the previous learn-to-construct model must learn not only the sub-route distribution $\prod_{t=1}^{l} p_\theta\left(r_t|s_t\right)$, but also the more complex and detailed sequences distribution inside the sub-route $\prod_{t=1}^{l} \prod_{i=1}^{l_t} p_\beta\left(\pi_i|r_t, \pi_{t,0:i-1}\right)$. In contrast, our TAM focus on learning the sub-route distribution. The sequence distribution inside a sub-route is handled by traditional or learned heuristics in our TAM. Therefore, 1) focusing on learning sub-route distribution makes our TAM much easy to train and learn; 2) the traditional heuristics could perform well on arbitrary small-scale TSPs and VRPs w.r.t any distributions, which could improve the generalization of our TAM to large-scale VRPs or distribution-shifted VRPs.

**2) The global mask function technique**

According to Equations 33, when generalizing from small-scale VRPs to large-scale VRPs, route number $l$ is one of the key variables. However, previous policies without global constraints cannot learn the possible range of optimal vehicle number in large-scale VRPs, which could generate a lot of vehicles instead. This is also one of the reasons behind the generalization failure of AM. For example, we found the AM method could generate 337 vehicles while our TAM with global mask function just generates 34 vehicles, shown in Figure 6. This means our global mask function helps encode maximum vehicle number when learning $\prod_{t=1}^{l} p_\theta\left(r_t|s_t\right)$ in our TAM. In addition, the pre-defined maximum vehicle number can help estimate the minimum vehicle number beforehand, which could improve the generalization of our TAM, about 5.8% for CVRP 2000.

**3) The proposed reward and RL training method** As above mentioned, both the proposed generating sub-route technique and the global mask function could help TAM generalize to large-scale VRPs in a zero-shot way. The third technique, the proposed reward and RL training method, makes training the two-stage dividing model possible. Our RL training method could accelerate the training process by using learned TSP model and parallel computing in GPU. Our reward makes the sequence-to-sequence policy in Equation 4 insensitive to the order of nodes inside each sub-route. Using this reward, we achieve generating sub-routes sequence rather than nodes sequence to improve the generalization of our TAM, about 2% for CVRP 2000.

## A.5 RELATED WORK

VRPs are usually solved by exact methods, traditional heuristics, and data-driven methods. The exact methods are time-consuming and incapable of solving large-scale VRPs in a reasonable time. Therefore, we mainly focus on the traditional heuristics and data-driven methods.

Traditional methods design heuristics based on domain knowledge to obtain acceptable solutions within a limited search time (Toth & Vigo, 2002b). Local search (Aarts et al., 2003; Voudouris et al., 2010), Genetic algorithms(Sivanandam & Deepa, 2008), Ant colony methods (Dorigo et al., 2006), Large neighborhood search(Pisinger & Ropke, 2010), and Tabu search (Taillard et al., 2001) are widely used. They usually adopt a construction, destruction, and improvement pattern (Golden et al., 2008). They first generate an original feasible solution by a constructor, and then destroy the current solution by a destructor, and then improve the current solution by an improvement operator (Ropke & Pisinger, 2006). The process is taken iteratively until we obtain an acceptable solution. The pattern is also used in solvers like Ortools (Perron & Furnon), LKH3 (Helsgaun, 2017), HGS (Vidal, 2022; Vidal et al., 2012), FILO (Accorsi & Vigo, 2021), SISRs (Christiaens & Vanden Berghe, 2020), and HILS (Subramanian et al., 2013). However, due to extremely large search space, the traditional heuristics need massive iterations to obtain acceptable solutions.

Data-driven methods could construct good VRP solutions directly without iterations (**learn-to-construct method**). Vinyals et al. first proposed using a learned heuristic with pointer network (PN) architecture to solve TSPs in real-time (Vinyals et al., 2015). The PN is trained using supervised learning and then generates the TSP solution as a sequence. From there, several improvements are witnessed. Bello et al. replace the supervised learning with policy-based RL to train PN-based heuristic (Bello et al., 2016). Nazari et al. extended RL-based PN to VRPs (Nazari et al., 2018). They use attention layer to replace LSTM in the encoder of PN. Kool et al. proposed using a Transformer as encoder in PN. Their attention model (AM) is successfully applied in several combinatorial optimization problems, such as TSP, VRP, Orienteering Problem(Kool et al., 2019). Their work obtains close to optimal results for VRP 100. Recently, Kool et al. proposed using dynamic programming to predict the heat-map of edges and then generate VRP solutions from it (Kool et al., 2021). Based on Kool's attention model, Kwon et al. observed the symmetry of VRP solutions and proposed POMO to improve the performance of AM. The learn-to-construct ideas are also applied in other variant VRPs and get promising results (Delarue et al., 2020; Peng et al., 2020; Falkner & Schmidt-Thieme, 2020; Xin et al., 2020). However, due to the difficulty of training the model on large-scale VRPs, generating solutions for VRPs with over 400 nodes are still challenging (Fu et al., 2021; Joshi et al., 2020; Ma et al., 2019). Some researchers attempted to generalize the learned model on small-scale VRPs to solve large-scale VRPs. For instance, Joshi et al. observed the difficulty to generalize the learned heuristics to larger instances (Joshi et al., 2020). They investigate how to generalize the learned TSP models to TSP 200 by choosing appropriate architecture and training methods. Their results show commonly used autoregressive decoding and RL could improve the generalization of the learned model, which are also used in our method. **Different from previous works**, we propose a novel method by taking the advantages of traditional heuristics, data-driven heuristics, and global information to generalize the learned heuristics to solve VRPs with over 1000 nodes in real-time.

Besides constructing VRPs solutions directly, other works have considered **learn-to-search** in exact methods (Gasse et al., 2019) and traditional heuristics (Lu et al., 2019; Khalil et al., 2017; Chen & Tian, 2019; Hottung & Tierney, 2019; Xin et al., 2021; Chen et al., 2020), such as learning better large neighborhood search heuristics (Hottung & Tierney, 2019; Chen et al., 2020), designing better constructors, destructors, and improvement operators (Lu et al., 2019; Khalil et al., 2017) to accelerate the search process. The results from previous works show the learn-to-search method could find better solutions for VRPs than traditional heuristics (Lu et al., 2019). However, the iterative search is still necessary.

**Decomposition technique** has been widely used in traditional heuristics (Queiroga et al., 2021; Zhang et al., 2021; Bosman & Poutré, 2006; Ventresca et al., 2013). The technique usually divides original VPRs by clustering method and then solves sub-problems in each cluster to obtain the final solutions (Cordeau et al., 2002; Vidal et al., 2013; Fisher & Jaikumar, 1981; Miranda-Bront et al., 2017). The cluster-based method is applied to solve CVRP (Xiao et al., 2019), PDP (Wang et al., 2020), VRPTW (Qi et al., 2012), and multi-depot VRP (Fan et al., 2021; Zhang et al., 2021; Dondo & Cerdá, 2007). However, the cluster-based method has two disadvantages: it is hard to

consider some common constraints and requires effortful human-designed heuristics. It is also hard to determine the optimal clustering number. Therefore, the clustering results could be infeasible or far from optimal solutions. **Different from previous works**, we propose using the RL-based attention model to split a large-scale VRP into several TSPs while satisfying the complex constraints. Recently, the decomposition ideas are also introduced to data-driven heuristics (especially learn-to-search heuristics) for solving larger instances (Fu et al., 2021), such as divide and conquer networks for TSP (Nowak et al., 2018) and learn-to-delegate method for VRP(Li et al., 2021). The divide and conquer network adopts a dynamic programming strategy to recursively split and merge the partial solutions to improve TSP solution. The learn-to-delegate method utilizes traditional solver like LKH3 as sub-problem solver to merge or improve the sub-routes recursively, which is inspired by POPMUSIC framework(Alvim & Taillard, 2013; Lalla-Ruiz & Voss, 2016; Taillard & Helsgaun, 2019). They both belong to learn-to-search methods. **Different from previous works**, our TAM contributes to generalize learn-to-construct heuristics in zero-shot way, which decomposes VRP as independent sub-routes and just call sub-problem solver once in parallel. Besides, our work could easily encode other VRP constraints by changing mask functions, which is more difficult for learn-to-search method. In following sections, we will focus on learn-to-construct method.

### A.6 MORE DISCUSSION

Analyzing our results, we found that all three proposed techniques matter for the generalization of TAM. 1) **Generating sub-route sequence and optimizing in parallel** help us take advantages of both data-driven and traditional heuristics. Comparing TAM-AM in Table 1 and TAM in Table 5, we found that our two-stage technique could improve TAM's generalization ability significantly, about 15% for CVRP 2000. By dividing the large-scale VRP into several independent small-scale sub-problems, we make solving the large-scale VRP in parallel and real-time possible. It should be noted that the small-scale sub-routes could be part of the routes, rather than complete routes like TSP. In this way, we could extend our TAM to other variants of VRP. 2) **The proposed reward and training method** help the learned first-stage model invariant to the order of nodes inside each sub-route. Using this reward, we improve the generalization of our TAM, about 2% for CVRP 2000 (Table 4). This reward also makes our TAM focus on finding good sub-routes split, rather than finding a good sequence for each sub-problem. 3) **The global mask function technique** helps our TAM estimate the minimum vehicle number beforehand. Therefore, TAM could learn to reserve enough capacity for the demands at the end of the sequence for large-scale VRPs, which could improve the generalization of our TAM, about 5.8% for CVRP 2000 (Table 4).

**Limitations:** although our TAM is promising for solving large-scale VRPs in real-time with its generalization ability, it still has some limitations: 1) If we want to generalize to even larger VRPs such as CVRP 100000, we may need to train our model on larger VRPs directly, such as CVRP 400 and 1000. Due to the Transformer architecture in the Encoder, our TAM consumes a lot of memory and computation resources when training on large-scale VRPs directly. 2) Our TAM outperforms learn-to-construct methods and is competitive with some traditional heuristics on large-scale VPRs in real-time and zero-shot generalization settings. However, when it comes to solving large-scale VRP with long searching time (such as days and hours), outperforming the SOTA traditional heuristics such as HGS-CVRP is still very challenging.

In the future work, we will focus on the following questions: 1) could we generalize our TAM to other constraints without retraining our model? 2) could we combine the advantages of both our TAM and learn-to-search methods to outperform SOTA heuristics on larger VRPs with a long searching time? 3) could we train our TAM on large-scale VRPs like CVRP 400 and 1000 to generalize on even larger VRPs, such as CVRP 100000? 4) could we embed more abstract forms of local and global constraints in our TAM? 5) could we generalize our method to other combinatorial problems?

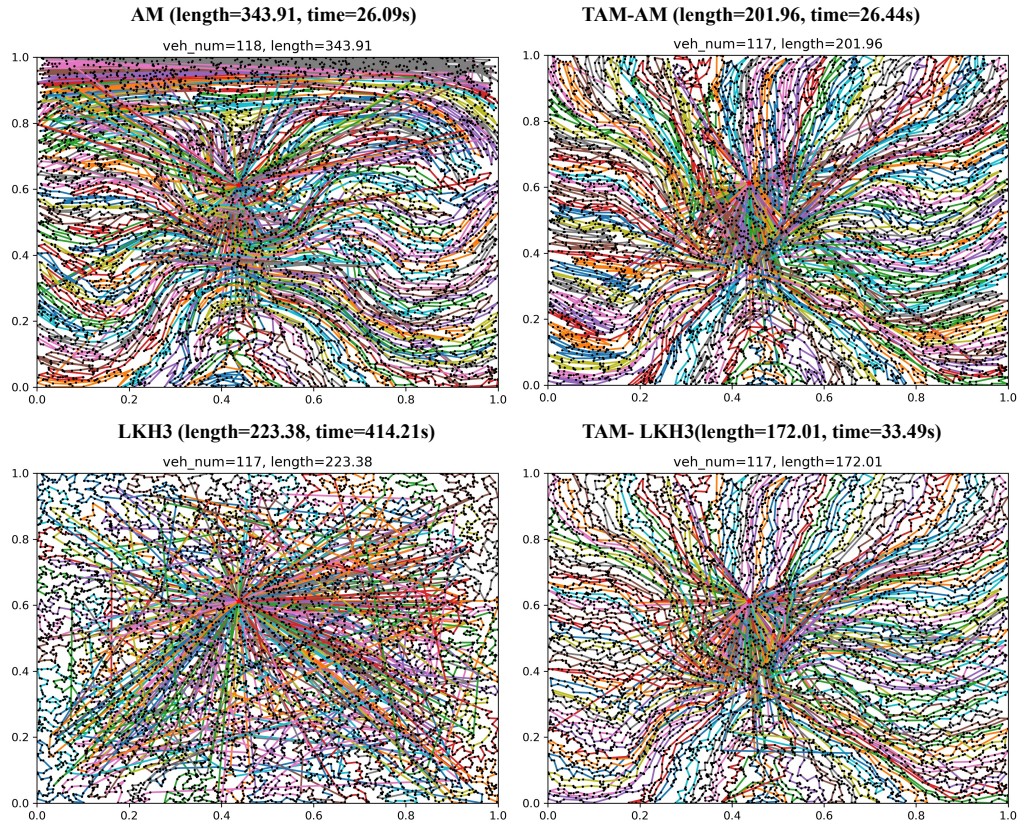

Figure 7: Example results on real-world VRP 7000. *Top left:* Data-driven Attention Model (**AM**). *Bottom left:* **LKH3**. *Top right:* Our TAM with Attention Model as the second-stage solver (**TAM-AM**). *Bottom right:* Our TAM with LKH3 as the second-stage solver (**TAM-LKH3**).

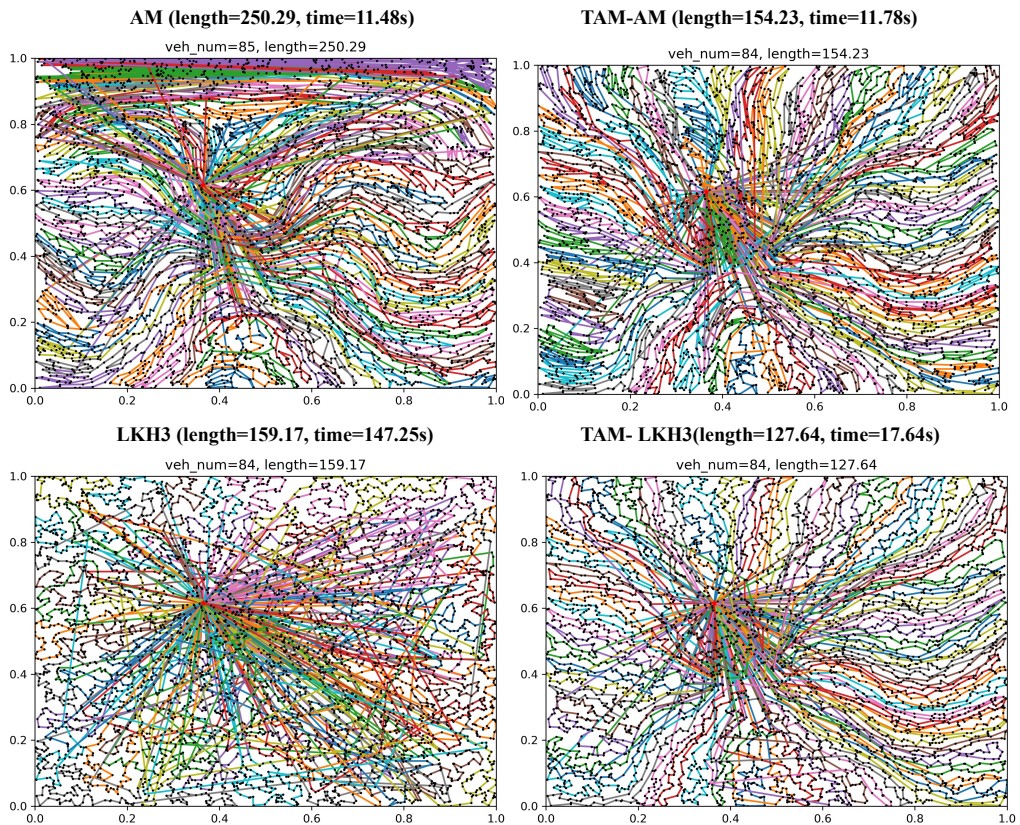

Figure 8: Example results on real-world VRP 5000. *Top left:* Data-driven Attention Model (**AM**). *Bottom left:* **LKH3**. *Top right:* Our TAM with Attention Model as the second-stage solver (**TAM-AM**). *Bottom right:* Our TAM with LKH3 as the second-stage solver (**TAM-LKH3**).

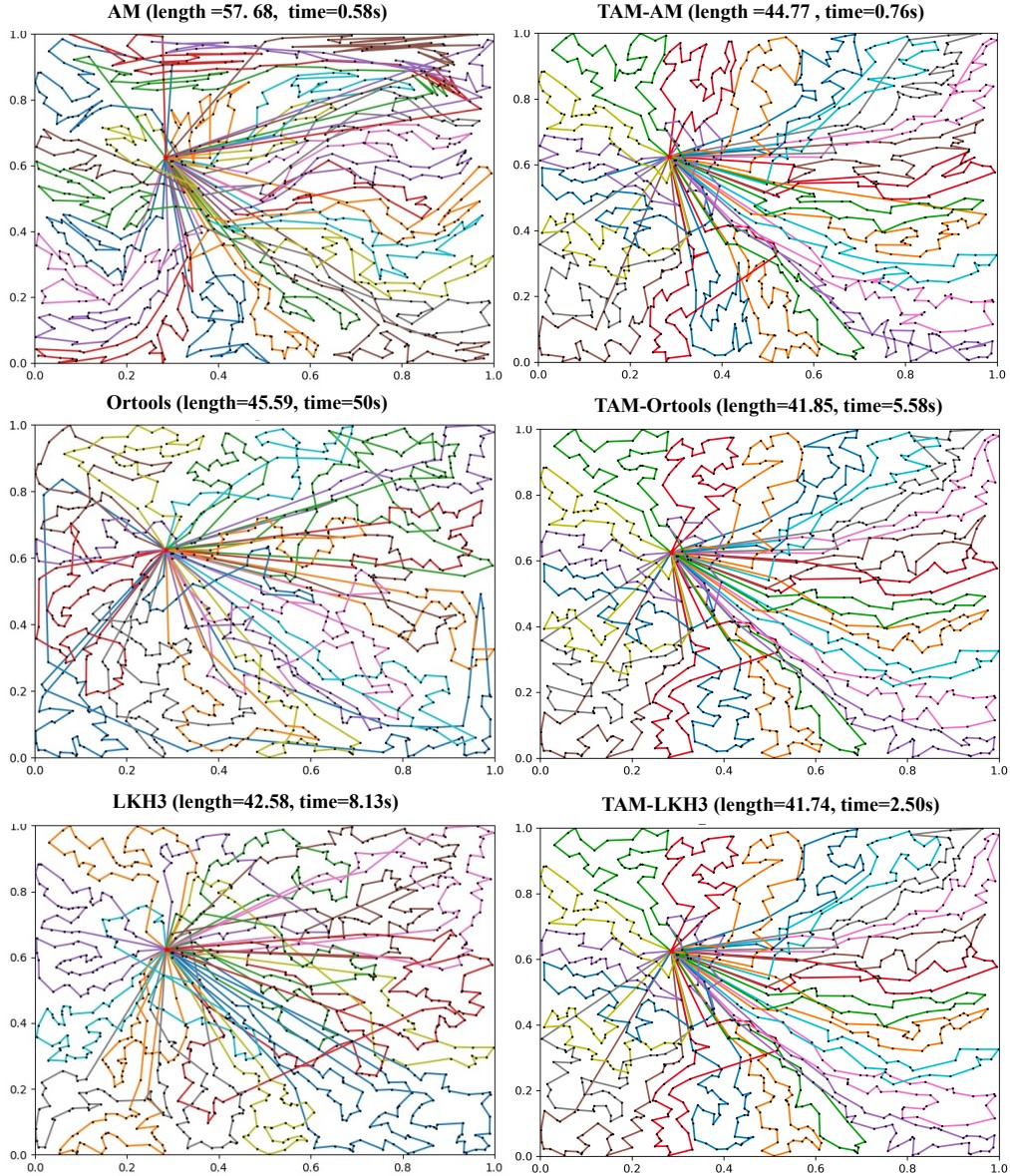

Figure 9: Example results on VRP 1000. *Top left:* Data-driven Attention Model (**AM**). *Medium left:* **Ortools**. *Bottom left:* **LKH3**. *Top right:* Our TAM with Attention Model as the second-stage solver (**TAM-AM**). *Medium right:* Our TAM with Ortools as the second-stage solver (**TAM-Ortools**). *Bottom right:* Our TAM with LKH3 as the second-stage solver (**TAM-LKH3**).

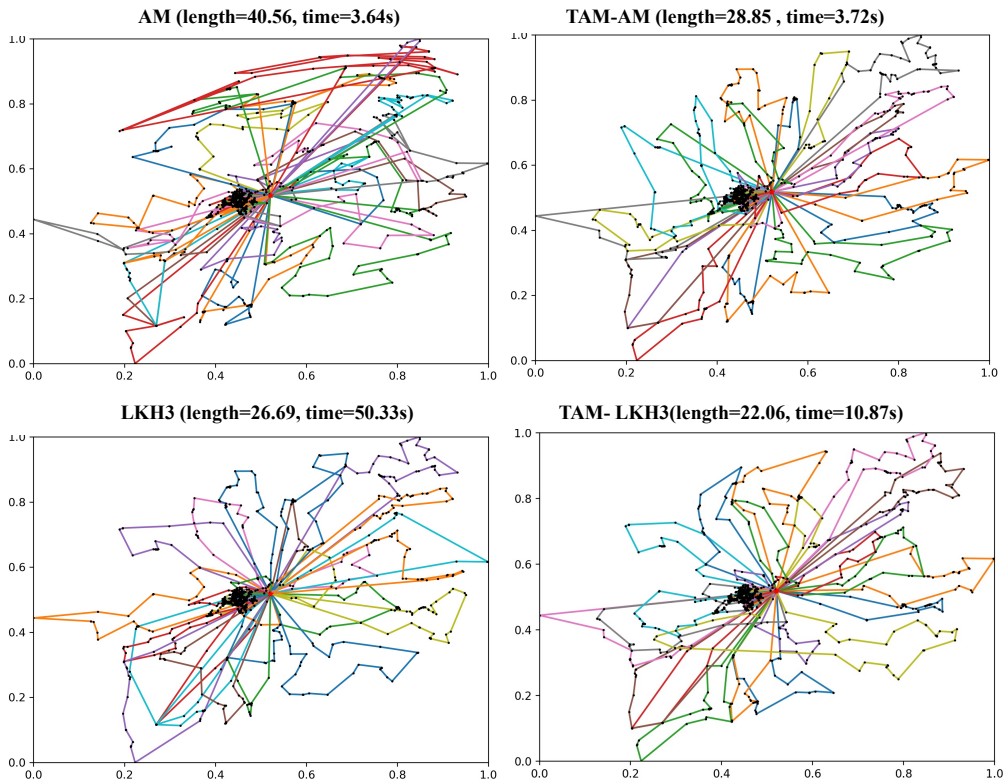

Figure 10: Example results on real-world VRP 1299. *Top left:* Data-driven Attention Model (**AM**). *Bottom left:* **LKH3**. *Top right:* Our TAM with Attention Model as the second-stage solver (**TAM-AM**). *Bottom right:* Our TAM with LKH3 as the second-stage solver (**TAM-LKH3**).

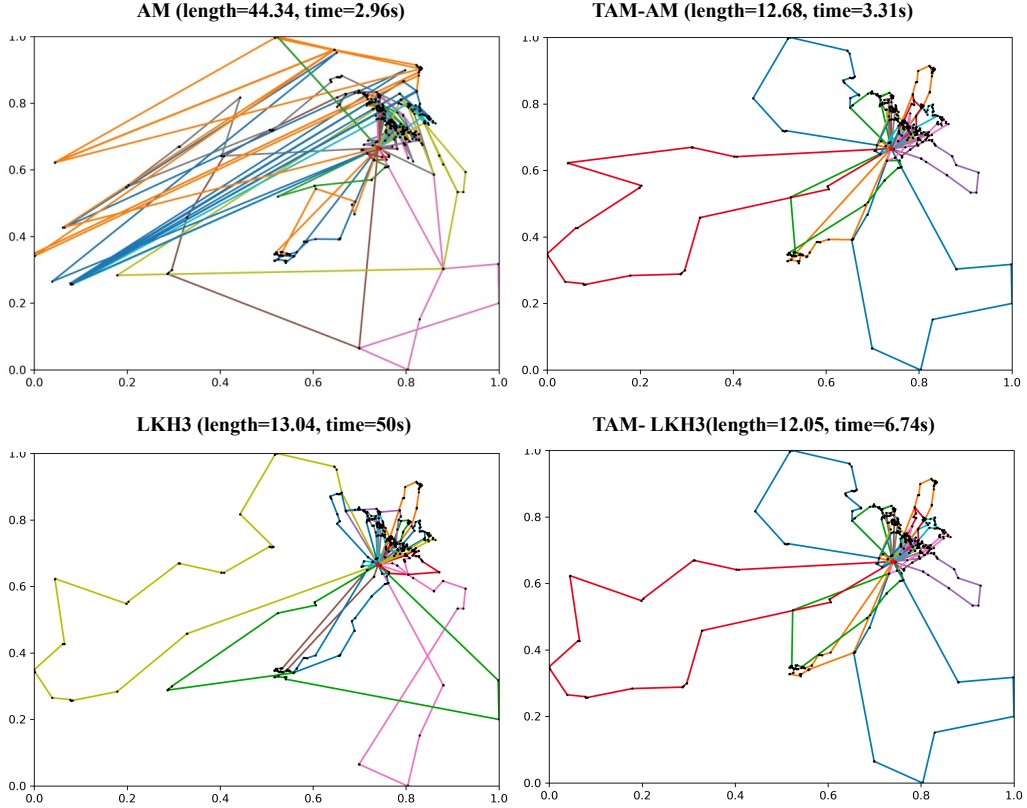

Figure 11: Example results on real-world VRP 963. *Top left:* Data-driven Attention Model (**AM**). *Bottom left:* **LKH3**. *Top right:* Our TAM with Attention Model as the second-stage solver (**TAM-AM**). *Bottom right:* Our TAM with LKH3 as the second-stage solver (**TAM-LKH3**).

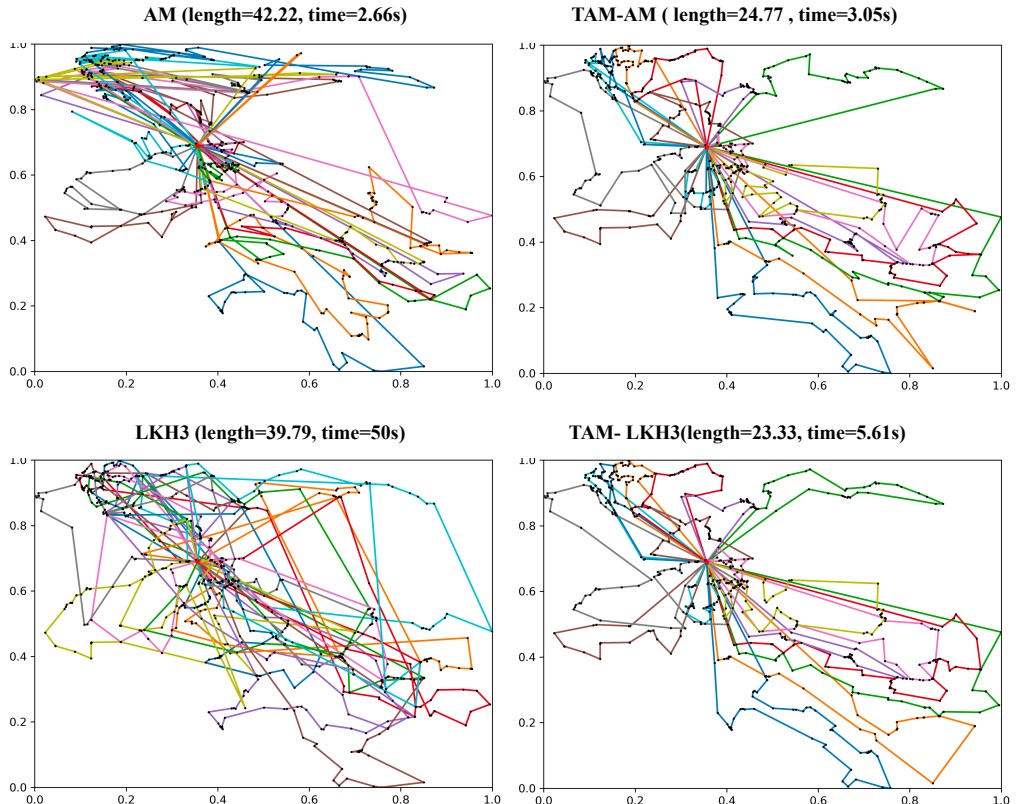

Figure 12: Example results on real-world VRP 864. *Top left:* Data-driven Attention Model (**AM**). *Bottom left:* **LKH3**. *Top right:* Our TAM with Attention Model as the second-stage solver (**TAM-AM**). *Bottom right:* Our TAM with LKH3 as the second-stage solver (**TAM-LKH3**).

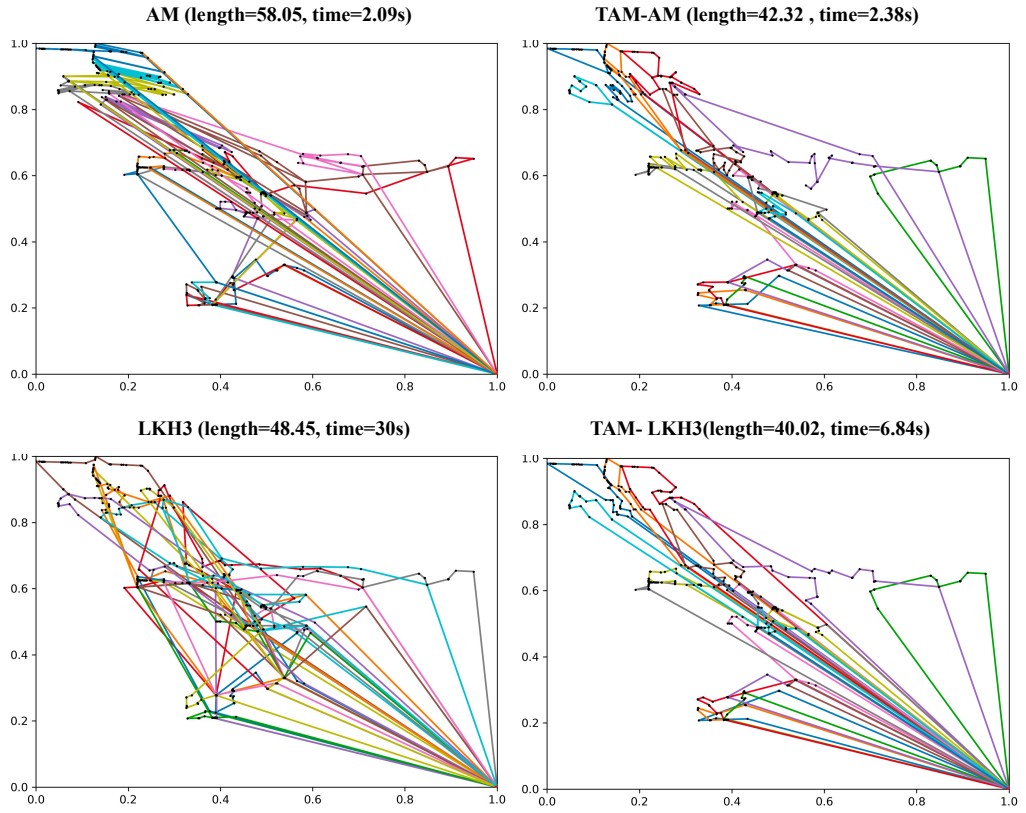

Figure 13: Example results on real-world VRP 680. *Top left:* Data-driven Attention Model (**AM**). *Bottom left:* **LKH3**. *Top right:* Our TAM with Attention Model as the second-stage solver (**TAM-AM**). *Bottom right:* Our TAM with LKH3 as the second-stage solver (**TAM-LKH3**).

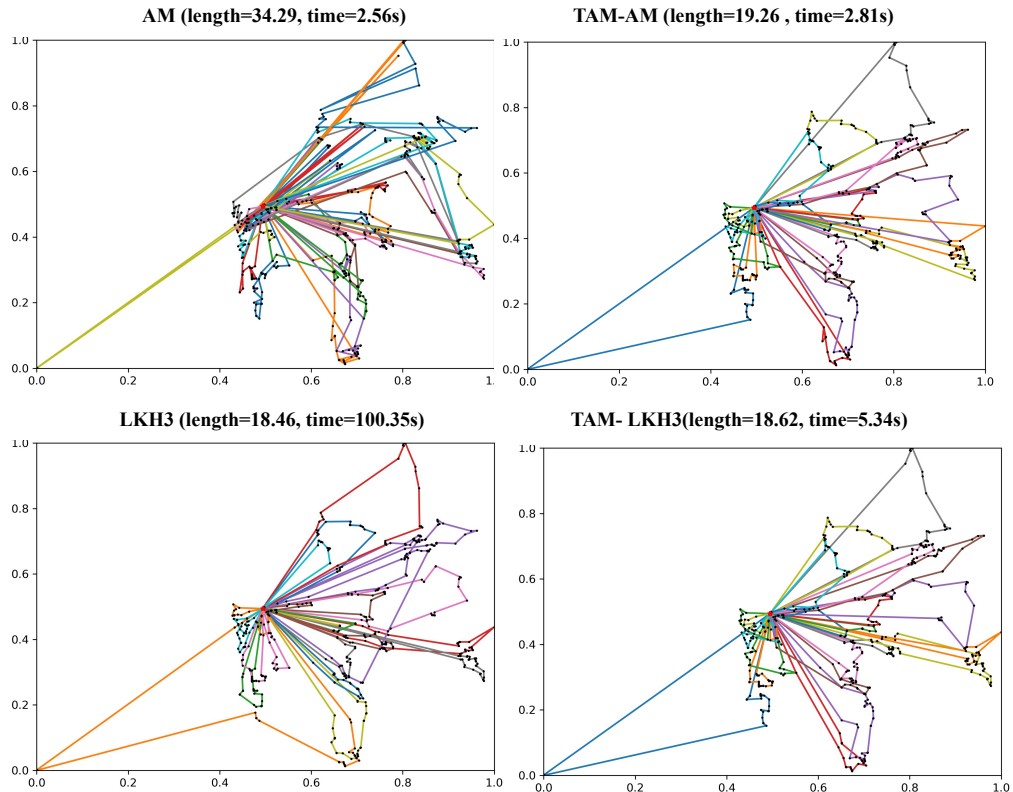

Figure 14: Example results on real-world VRP 821. *Top left:* Data-driven Attention Model (**AM**). *Bottom left:* **LKH3**. *Top right:* Our TAM with Attention Model as the second-stage solver (**TAM-AM**). *Bottom right:* Our TAM with LKH3 as the second-stage solver (**TAM-LKH3**).

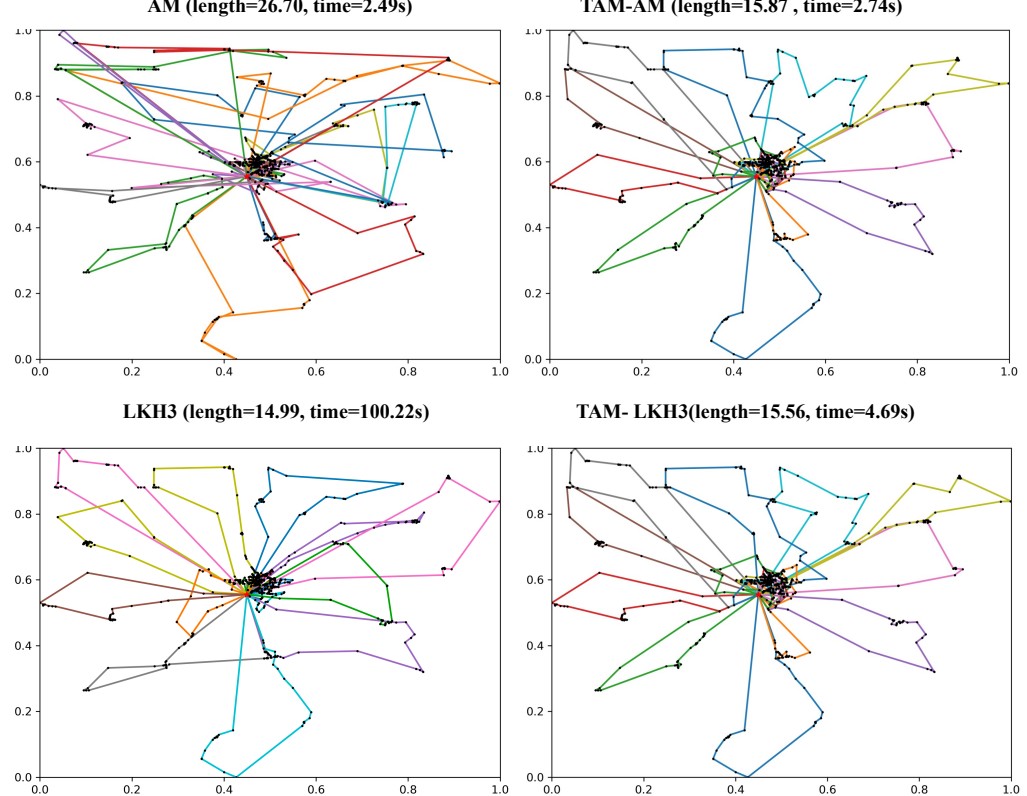

Figure 15: Example results on real-world VRP 817. *Top left:* Data-driven Attention Model (**AM**). *Bottom left:* **LKH3**. *Top right:* Our TAM with Attention Model as the second-stage solver (**TAM-AM**). *Bottom right:* Our TAM with LKH3 as the second-stage solver (**TAM-LKH3**).

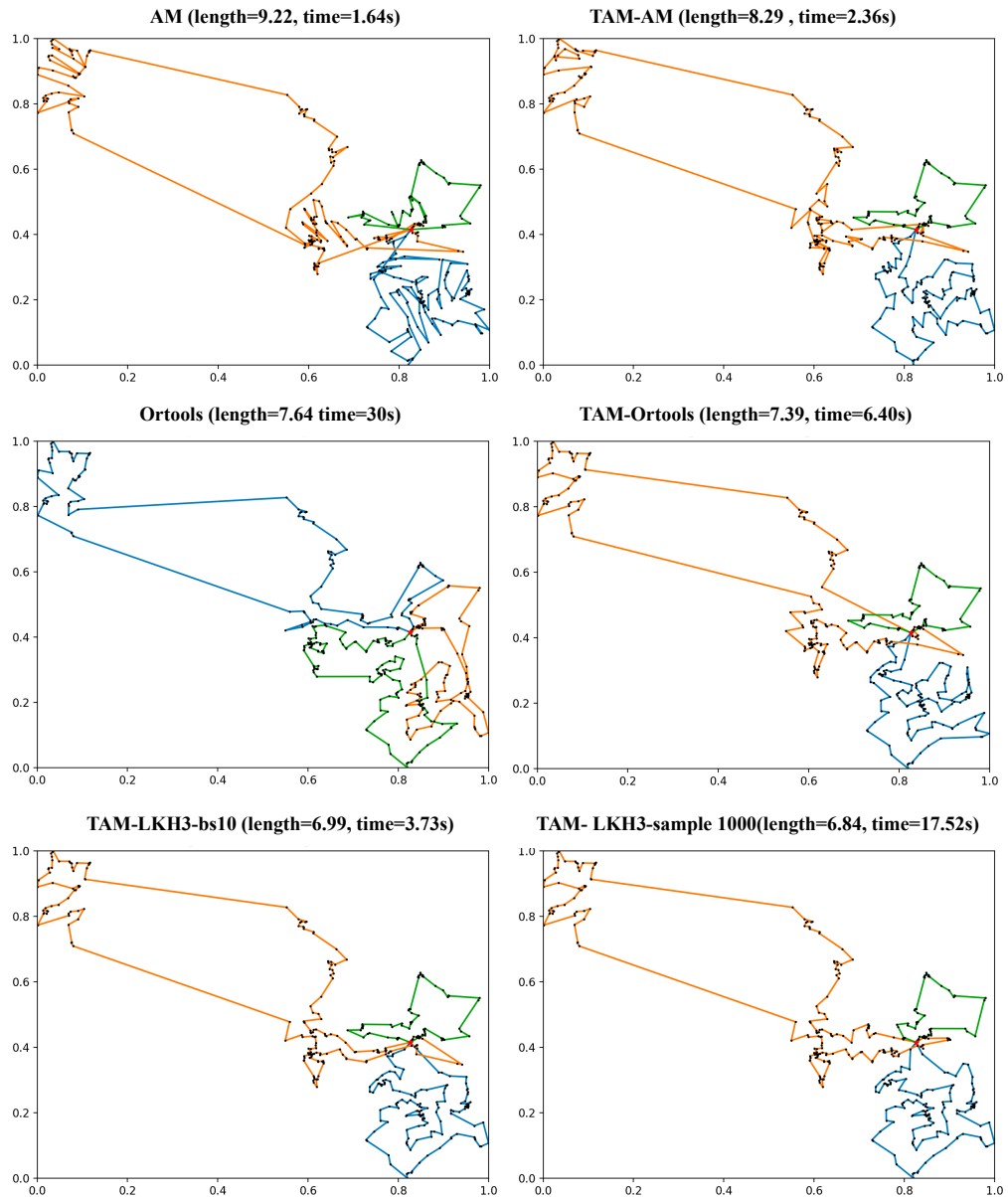

Figure 16: Example results on real-world VRP 388. *Top left:* Data-driven Attention Model (**AM**). *Medium left:* **Ortools**. *Bottom left:* TAM (beam size =10) with LKH3 as the second-stage solver (**TAM-LKH3**). *Top right:* Our TAM with Attention Model as the second-stage solver (**TAM-AM**). *Medium right:* Our TAM with Ortools as the second-stage solver (**TAM-Ortools**). *Bottom right:* TAM (sample size =1000) with LKH3 as the second-stage solver (**TAM-LKH3-sample 1000**).

