# OpenReview forum: "Generalize Learned Heuristics to Solve Large-scale Vehicle Routing Problems in Real-time"
_ICLR.cc/2023/Conference — ICLR 2023 poster_

### Official Review · Reviewer_6VLJ · 2022-10-23

**Confidence:** 4
**Correctness:** 3
**Technical Novelty And Significance:** 2
**Empirical Novelty And Significance:** 3
**Recommendation:** 6

**Clarity, Quality, Novelty And Reproducibility:**

- The general idea is well explained at a higher level in Fig.1 and Fig.2. The introduction of MDP beyond the existing AM method (Kool et al. 2019) is given and explained in Eq.1 to Eq.4. Technical contributions on rewards and mask functions a explained in Eq.5 to Eq.7. In my opinion, authors clearly explained the idea. On possible drawback is the training part is almost included only in Appendix, although the authors followed as standard learning framework on RL.

- The quality of the proposed method is reasonable, i.e., it is based on well explained concepts in the paper and standard expreimental evaluations are done.

- The reproducability is now not easily evaluated for me. One reason is the proposed experiments used many related methods (e.g., AM, LKH3, TAM, ...). Another reason is no explict explanations on algorihtms (e.g., pseudo-codes). This could be improved by providing some supplemental materials.

**Strength And Weaknesses:**

[Strength]
- TAM can be attached with any sub-problems solvers (existing heuristics like LKH3 and data-driven methods like AM).
- TAM possibly accelerates existing heuristics (e.g., AM v.s. TAM+AM) by improving their solution qualities.
- Ablation studies for reward and mask functions are meaningful (i.e., designing rewards is an essential step for learning-based methods).

[Weakness]
- Technical contributions are a bit incremental.

**Summary Of The Paper:**

The paper proposes a two-stage attention-based method (TAM) to solve large-scale routing problems.

Although existing learning-based solvers suffer from their scalability (e.g., # of nodes is typically less than a thousand), TAM is trained to split the large input instance into small sub-problems with keeping global constraints. Importantly, TAM can be attached to existing solvers to improve solution qualities. The technical contribution is by attention-based problem splitting, and then sub-problems can be efficiently solved by existing heuristics or learned data-driven methods.

Experimental results are reported to support and discuss TAM. Many experimental results support the performance of the proposed approach emperically.

**Summary Of The Review:**

The paper proposes a two-stage attention-based method (TAM) to solve large-scale routing problems. In recent years, simliar problems have been investigated by several researchers, but the proposed paper seems to succeed to solve larger instances. The proposed method seemes to be technically sound, a bit incremental improvement, but a large experimental advantages can be observed by experiments on synthethic and real instance data. Importantly the advantages seems to be large, and the result (i.e., solutions of routing problems) could be much better than existing approaches.

Based on these observations, I feel that the proposed paper is above the baseline.

---

> ### Author Response · Authors · 2022-11-18
> **Response to Reviewer 6VLJ**
>
> We really appreciate your recognition and valuable comments. We agree with you, also noted by Reviewer o9TS and Reviewer cdJ3, that the idea of TAM is not complex but effective. We also agree with you that the existing learning-based solvers suffer from scalability problems, but TAM is trained to split the large input instance into small sub-problems while keeping global constraints. In the following part,  we have tried our best to address your main concerns.
>
> **Technical contributions**
>
> Our main contribution is generalizing the TAM learned on small-scale VRPs to solve large-scale VRPs with over 5000 nodes in seconds.  To achieve this goal, we propose three techniques to improve the generalization of TAM:
>
> 1) Generating sub-route sequence and optimizing in parallel
>
> The previous learn-to-construct model must learn not only the sub-route distribution $\prod_{t=1}^{l} p_{\theta}\left(r_{t} | s_t\right)$, but also the more complex and detailed sequences distribution inside the sub-route $\prod_{t=1}^{l}\prod_{i=1}^{l_t} p_{\beta}\left(\pi_{i} | r_t, \pi_{t, 0:i-1}\right)$.   In contrast, our TAM focus on learning the sub-route distribution.  The sequence distribution inside a sub-route is handled by traditional or learned heuristics in our TAM. Therefore, 1) focusing on learning sub-route distribution makes our TAM much easy to train and learn;  2) the traditional heuristics could perform well on arbitrary small-scale TSPs and VRPs w.r.t any distributions, which could improve the generalization of our TAM to large-scale VRPs or distribution-shifted VRPs.
>
> 2) The global mask function technique
>
> According to Equations 3, when generalizing from small-scale VRPs to large-scale VRPs, route number $l$ is one of the key variables. However, previous policies without global constraints cannot learn the possible range of optimal vehicle number in large-scale VRPs,  which could generate a lot of vehicles instead. This is also one of the reasons behind the generalization failure of AM. For example, we found the AM method could generate 337 vehicles while our TAM with global mask function just generates 34 vehicles, shown in Figure 6 of Appendix. This means our global mask function helps  encode maximum vehicle number when learning $\prod_{t=1}^{l} p_{\theta}\left(r_{t} | s_t\right)$in our TAM. In addition, the pre-defined maximum vehicle number can help estimate the minimum vehicle number beforehand, which could improve the generalization of our TAM, about 5.8\% for CVRP 2000.
>
> 3) The proposed reward and RL training method
>
> As above mentioned, both the proposed generating sub-route technique and the global mask function could help TAM generalize to large-scale VRPs in a zero-shot way.  The third technique, the proposed reward and RL training method, makes training the two-stage dividing model possible.  Our RL training method could accelerate the training process by using learned TSP model and  parallel computing in GPU. Our reward makes the sequence-to-sequence policy in Equation 4 insensitive to the order of nodes inside each sub-route. Using this reward, we achieve generating sub-routes sequence rather than nodes sequence to improve the generalization of our TAM, about 2\% for CVRP 2000.
>
> **Reproducibility**
>
> As pointed, our TAM with the proposed three generalization techniques (generating subroute sequence, new reward, and global mask function) is not complex but effective. It can be easily implemented based on seq2seq model (such as Attention Model and GNN) and open-sourced solver (such as LKH3 and Ortoools).  The details of the training algorithm implementation are shown in  Algorithm 1 in Appendix A.3.1. There are three implementation differences between TAM and traditional RL method: 1) training an Attention Model for solving TSP in advance, 2) padding all sub-TSPs to the same length, 3) calculate the reward with Equation 18.  As suggested, we add the implementation details in Algorithm 2 and Algorithm 3 in Appendix of the revised version for the inference algorithm to improve reproducibility. If the paper is accepted. we will also release an implementation once getting necessary permission.

---

### Official Review · Reviewer_7d5S · 2022-10-24

**Confidence:** 3
**Correctness:** 3
**Technical Novelty And Significance:** 2
**Empirical Novelty And Significance:** 2
**Recommendation:** 5

**Clarity, Quality, Novelty And Reproducibility:**


<Methodology>

1. It is understandable that the dimension of the action that selects a cluster is large, so the current study proposes to use a sequence-to-sequence model. However, it is not clearly explained how to reduce the influence on the order. The factorial of the number of nodes generates the same reward. Isn't this redundancy affecting learning? In other words, shouldn't the attribute of permutation invariance be assigned to the learned policy?


2. In general, when decomposing nodes, clustering-based methodologies such as k-means seem to be used a lot. Compared to these methodologies, what are the advantages of decomposing-based methodologies based on sequence to sequence model? Even when the number of clustering increases, is it possible to effectively cluster by reflecting the relationships between nodes?

3. In Equation 4, why is the summation of the product up to n + l? What is n?




<Experiments>

1. Comparison with other decomposing strategies is necessary. Comparing the proposed method with a non-decomposing strategy is not fair. Because the current study proposes an effective decomposing strategy, it should compare with other effective decomposing strategies that can solve the large-scale CVRP. In addition, the performance criterion should be the optimality gap measuring the difference between the produced solution by the proposed method and the true optimum. Because many papers share the best-known solutions for large-scale CVRPs, the study should use this optimal solution as the oracle reference. The current experiment results look like a just ablation study showing the effectiveness of using the decomposing strategy.


2. At what size did AM and POMO learn? Isn't it an unfair comparison to learn them on small size and then solve them on a large size? These models are not intended to achieve transferability over size. If that is the case, wouldn't it be better to compare the performance with other methods designed to achieve size transferability?


**Strength And Weaknesses:**

The decomposition strategy is a methodology that has been dealt with a lot on the OR side. A literature review of decomposition methods is required. Additionally, more recently, the NCO community has been using a decomposition strategy to deal with large-scale VRP issues. It is difficult to evaluate this paper without including these papers in the literature review and comparing them with these methods.

The current study insists that employing decomposing strategy in a constructive way (i.e., constructive heuristics) is novel in that most decomposing strategies are employed in an iterative manner (i.e., improving heuristics). However, I think it's hard to claim novelty just by suggesting a way to find a solution in a different way. For it to be truly advantageous, it is when the quality of the solution derived by such a process approaches the optimum. From that point of view, I think the strategy of finding a good solution by repeatedly decomposing and solving sub-problems can be more effective.


**Summary Of The Paper:**

The current study, as a strategy to effectively solve large-scale CVRP, presents a strategy for generating TSP sub-problems by clustering nodes and simultaneously solving the generated TSP problems. In particular, the sequential node selection approach has been used to decompose nodes.

**Summary Of The Review:**

Although the proposed methodology is interesting, it is necessary to quantify how much more effective the proposed decomposition strategy is compared to other decomposition strategies.

---

> ### Author Response · Authors · 2022-11-18
> **Response to Reviewer 7d5S -Part I**
>
> We are extremely grateful for taking your time to review our paper.  We have carefully thought about all your comments. We have tried our best to address your main concerns in the following part.
>
> **Decomposition literature from OR field**
>
> We agree with you that the decomposition strategy (divide and conquer) is widely used in OR domain. But the combination of decomposition strategy and learning method (RL and deep learning) is less explored, which is the main focus of our paper. That's why we included several decomposition literatures from both OR and ML fields and reviewed them in the related works in Appendix A.4 of the original manuscript.  As suggested, we will include more decomposition-related literatures from OR field in the final version. The literatures are as follows:
>
> [1] Fan H, Zhang Y, Tian P, et al. Time-dependent multi-depot green vehicle routing problem with time windows considering temporal-spatial distance[J]. Computers \& Operations Research, 2021, 129: 105211.
>
> [2] Xiao J, Zhang T, Du J, et al. An evolutionary multiobjective route grouping-based heuristic algorithm for large-scale capacitated vehicle routing problems[J]. IEEE transactions on cybernetics, 2019, 51(8): 4173-4186.
>
> [3] Zhang Y, Mei Y, Huang S, et al. A route clustering and search heuristic for large-scale multidepot-capacitated arc routing problem[J]. IEEE Transactions on Cybernetics, 2021.
>
> [4] Miranda-Bront J J, Curcio B, Méndez-Díaz I, et al. A cluster-first route-second approach for the swap body vehicle routing problem[J]. Annals of Operations Research, 2017, 253(2): 935-956.
>
> [5] Wang Y, Lei L, Zhang D, et al. Towards delivery-as-a-service: Effective neighborhood search strategies for integrated delivery optimization of E-commerce and static O2O parcels[J]. Transportation Research Part B: Methodological, 2020, 139: 38-63.
>
> [6] Qi M, Lin W H, Li N, et al. A spatiotemporal partitioning approach for large-scale vehicle routing problems with time windows[J]. Transportation Research Part E: Logistics and Transportation Review, 2012, 48(1): 248-257.
>
> [7] Ouyang Y. Design of vehicle routing zones for large-scale distribution systems[J]. Transportation Research Part B: Methodological, 2007, 41(10): 1079-1093.
>
> [8] Thangiah S R, Salhi S. Genetic clustering: an adaptive heuristic for the multidepot vehicle routing problem[J]. Applied Artificial Intelligence, 2001, 15(4): 361-383.
>
> [9] Dondo R, Cerdá J. A cluster-based optimization approach for the multi-depot heterogeneous fleet vehicle routing problem with time windows[J]. European journal of operational research, 2007, 176(3): 1478-1507.
>
> [10] Vidal T, Crainic T G, Gendreau M, et al. Heuristics for multi-attribute vehicle routing problems: A survey and synthesis[J]. European Journal of Operational Research, 2013, 231(1): 1-21.
>
> [11] Fisher M L, Jaikumar R. A generalized assignment heuristic for vehicle routing[J]. Networks, 1981, 11(2): 109-124.
>
> [12] Cordeau J F, Gendreau M, Laporte G, et al. A guide to vehicle routing heuristics[J]. Journal of the Operational Research society, 2002, 53(5): 512-522.
>
> [13] Alesiani F, Ermis G, Gkiotsalitis K. Constrained Clustering for the Capacitated Vehicle Routing Problem (CC-CVRP)[J]. Applied artificial intelligence, 2022, 36(1): 1995658
>
> **The meaning of learn-to-construct heuristics**
>
> We agree with you that it's hard to claim novelty just by suggesting a way to find a solution in a different way. But the logic and ideas behind the learn-to-construct method and learn-to-search method are totally different. Both methods have its own advantages and application scenarios. 1) The learn-to-search methods such as learn-to-improve search good solutions with massive iterations from a feasible initial solution. If given enough time, the learn-to-search method could always improve the current solutions. Therefore, the learn-to-search method is more suitable for applications sensitive to the solution quality but insensitive to the solving time. 2) The Learn-to-construct methods such as AM, POMO, and our TAM could generate good solutions in real-time directly without iterations. Therefore, the learn-to-construct method is suitable for online or real-time applications where the requirement for solving time is strong.  In fact, as mentioned in search strategy of Appendix A.3.2, TAM could also search better solutions by changing the beam size or sample size of the search strategy. The search size controls the trade-off between the solution quality and solving time. In addition, we could also take the advantages of both TAM and learn-to-search method by using TAM to quickly generate a good initial solution and then improving the solution using the learn-to-search method to balance the solution quality and solving time.

---

> > ### Author Response · Authors · 2022-11-18
> > **Response to Reviewer 7d5S -Part II**
> >
> > > **Question 1: It is understandable that the dimension of the action that selects a cluster is large, so the current study proposes to use a sequence-to-sequence model.  However, it is not clearly explained how to reduce the influence on the order. The factorial of the number of nodes generates the same reward. Isn't this redundancy affecting learning? In other words, shouldn't the attribute of permutation invariance be assigned to the learned policy?**
> >
> > As you pointed, we reformulate the original set2set model as seq2seq model for two considerations:  1) avoid sampling all nodes inside a sub-route simultaneously; 2)  easily encode various constraints by mask function in seq2seq model.  We also agree that the learned policy should have permutation invariance to the input (the order of nodes) and output (the order of nodes inside a sub-route). To achieve this invariance to the output, we proprose using the optimal length of the sub-route as new reward. Because the optimal length is invariant to the order of TSP inputs, the reward is then invariant to the order of nodes inside the sub-route. To achieve the permutation invariance to the input, the positional encoding is removed from the multi-head attention model of the Encoder as AM. Due to the above formulation, our policy and reward is permutation invariance to the order of both input and output.
> >
> > > **Question 2: In general, when decomposing nodes, clustering-based methodologies such as k-means seem to be used a lot. Compared to these methodologies, what are the advantages of decomposing-based methodologies based on sequence to sequence model? Even when the number of clustering increases, is it possible to effectively cluster by reflecting the relationships between nodes?**
> >
> > Compared with clustering-based method, our TAM has two advantages:
> >
> > 1) TAM could easily encode various constraints in seq2seq model by mask function, such as capacity, maximum vehicle number, and time window. In contrast, it is hard for clustering-based method to consider some common constraints (like vehicle capacity, time window) and requires effortful human-designed heuristics. Therefore, the dividing results of the clustering-based method could be infeasible.
> >
> > 2) TAM could find the optimal vehicle number after training (clustering number in this paper). In contrast, the clustering-based method such as k-means should choose the optimal clustering number by human beforehand. Therefore, the results of clustering-based method could far from the optimal solutions.
> >
> > As the number of nodes increases, the number of vehicles or clusters also increases. The experiments on synthetic and real-world instances show our TAM trained on VRP 100 could generalize well to solve the large-scale VRPs with over 1000 nodes. This means our TAM could reflect potential relations between nodes and then find the optimal vehicle number. For instance, Figure 6 in Appendix shows the vehicle number of TAM is about 10 times less than that of AM, which is close to the optimal vehicle number.
> >
> > > **Question 3: In Equation 4, why is the summation of the product up to n + l? What is n?**
> >
> > As mentioned in Section 3, the $n$ is the number of customers, $l$ is the number of vehicles. If we reformulate Equation 2 as seq2seq model, the depot is visited by $l$ times and each node is visited once. Therefore, the conditional probability function is multiplied by $n + l$ times.

---

> > > ### Author Response · Authors · 2022-11-18
> > > **Response to Reviewer 7d5S - Part III**
> > >
> > > **Question 1 for experiments**
> > >
> > > **Comparison with decomposition method from OR fields** We compare TAM with LKH-3 and Ortools from OR fields rather than the decomposition method in the original version for the following considerations.
> > >
> > > 1) The cluster-based method has two disadvantages: it is hard to consider some common constraints (like vehicle capacity, time window) and requires effortful human-designed heuristics. It is also hard to determine the optimal clustering number. To this end, we propose using the RL-based model to split a large-scale VRP into several TSPs while satisfying the complex constraints.  Our TAM could learn good global split in real-time while considering common constraints such as capacity, maximum vehicle number, and time window by mask function.
> > >
> > > 2) This paper focuses on generalizing the learned heuristics to solve large-scale VRPs in real-time. Therefore, the generalization-ability and scalability matter more for our TAM. That's why we designed the current experiments and benchmarks. We found that generating sub-route sequence, new reward, and global mask function could improve the generalization of learned heuristics. The results show our TAM could significantly outperform both data-driven method and well-known heuristics from OR fields in real-time.
> > >
> > > 3) The decomposition strategy is often integrated into VRP solvers. Therefore, we chose well-known and open-sourced solvers such as LKH-3, Ortools, and HGS for comparison.
> > >
> > >
> > > **Optimal solutions** Following your suggestions, we also test our TAM on several all **52** instances with over 500 nodes on CVRPLIB.  The following Tables report the gap and solving time of several large instances. Please refer to the response to Reviewer cdJ3 for more results https://openreview.net/forum?id=6ZajpxqTlQ&noteId=12mlc2P6JO. The results show that our TAM-LKH3 could find significantly better results than AM and could find a better solution much faster than LKH3.  We also observed that the gap between our TAM and the best solution is about 5-10\% when generalizing to VRP with about 1000 nodes, and about 15-25\% when generalizing to VRP with over 3000-10000 nodes.  This is acceptable for real-time applications given that we limit the maximum solving time of our TAM as 60 seconds and the beam search size as 10 for real-time applications.  1) larger-scale VRP is much harder to solve; 2) the difference between large-scale VRP and VRP 100 (used for training TAM) enlarges as the nodes number increases.   If the optimal solutions are required for non-real-time applications, we should significantly improve the sample size of our TAM to find a better solution.
> > >
> > > Gap of CVRPLIB instances
> > > | Length        | No. Nodes | AM     | TAM-AM | LKH3   | TAM-LKH3 |
> > > |---------------|-----------|--------|--------|--------|----------|
> > > | X-n895-k37    | 895       | 36.84% | 14.32% | 29.72% | 9.39%    |
> > > | X-n916-k207   | 916       | 17.19% | 8.91%  | 10.34% | 8.87%    |
> > > | X-n936-k151   | 936       | 27.86% | 13.92% | 22.89% | 13.47%   |
> > > | X-n979-k58    | 979       | 30.41% | 8.23%  | 15.80% | 6.70%    |
> > > | X-n1001-k43   | 1001      | 18.36% | 12.99% | 13.79% | 10.64%   |
> > > | Leuven1-3001  | 3001      | 46.93% | 20.24% | 18.10% | 19.30%   |
> > > | Leuven2-4001  | 4001      | 53.31% | 38.57% | 22.14% | 15.88%   |
> > > | Antwerp1-6001 | 6001      | 39.30% | 24.90% | 24.20% | 24.01%   |
> > > | Antwerp2-7001 | 7001      | 50.32% | 33.20% | 31.09% | 22.55%   |
> > > | Ghent1-10001  | 10001     | 46.89% | 30.20% | -      | 29.53%   |
> > > | Ghent2-11001  | 11001     | 52.20% | 33.29% | -      | 23.65%   |
> > >
> > > Time of CVRPLIB instances
> > > | Time (s)               | No. Nodes | AM     | TAM-AM | LKH3    | TAM-LKH3 |
> > > |--------------------|-----------|--------|--------|---------|----------|
> > > |        X-n895-k37  | 895       | 2.64   | 2.58   | 4.23    | 3.91     |
> > > |        X-n916-k207 | 916       | 3.16   | 3.09   | 8.93    | 6.84     |
> > > | X-n936-k151        | 936       | 3.94   | 3.41   | 9.22    | 6.05     |
> > > | X-n979-k58         | 979       | 3.03   | 2.95   | 4.22    | 5.06     |
> > > | X-n1001-k43        | 1001      | 2.95   | 2.93   | 4.18    | 4.68     |
> > > | Leuven1-3001       | 3001      | 9.73   | 9.82   | 68.99   | 15.59    |
> > > | Leuven2-4001       | 4001      | 13.45  | 13.68  | 73.96   | 23.73    |
> > > | Antwerp1-6001      | 6001      | 12.81  | 12.98  | 596.21  | 24.91    |
> > > | Antwerp2-7001      | 7001      | 14.96  | 15.26  | 479.47  | 31.90    |
> > > | Ghent1-10001       | 10001     | 21.41  | 21.66  | -       | 37.26    |
> > > | Ghent2-11001       | 11001     | 38.55  | 37.56  | -       | 55.73    |

---

> > > > ### Author Response · Authors · 2022-11-18
> > > > **Response to Reviewer 7d5S - Part IV**
> > > >
> > > > > **Question 2 for experiments: At what size did AM and POMO learn? Isn't it an unfair comparison to learn them on small size and then solve them on a large size? These models are not intended to achieve transferability over size. If that is the case, wouldn't it be better to compare the performance with other methods designed to achieve size transferability?**
> > > >
> > > > AM and POMO are trained on the same dataset of VRP 100 as TAM and then generalized to solve large-scale VRPs. We agree with you that the transferability over size is very important for practical applications. In fact, the AM and POMO are intended to achieve scale generalization. For example, the generalization of AM are shown in Figure 5 of the AM paper [1]. It states that "we test generalization performance on different $n$ than trained for, which we plot in Figure 5 in terms of the relative optimality gap compared to Gurobi"  and "The models generalize when tested on different sizes, although quality degrades as the difference becomes bigger". That's also one of the main contributions of our TAM. To show the contribution, we choose the AM and POMO as the benchmarks. For a fair comparison, AM and POMO are given the same dataset and information as our TAM. If our TAM is trained on dataset of VRP100 while other methods are trained on the dataset of large-scale VRPs, it would be unfair for TAM because the AM and POMO are given additional information. In addition, to the best of our knowledge, TAM is the first learn-to-construct method that could generalize well from VRP 100 to solve VRP with over 5000 nodes in seconds. It's hard for us to find proper learn-to-construct methods that achieve size transferability. Given the above considerations, we choose AM and POMO as benchmarks of learn-to-construct methods. To test the generalization ability of TAM, we also choose the well-known open-sourced LKH-3 and Ortools as the baseline, because the traditional heuristics from OR fields have a good generalization on an arbitrary dataset.
> > > >
> > > > [1] Kool W, Van Hoof H, Welling M. Attention, learn to solve routing problems![J]. arXiv preprint arXiv:1803.08475, 2018.

---

> ### Author Response · Authors · 2022-12-08
> **Response to Reviewer 7d5S-More comparison with decomposing-based method**
>
> Following your suggestions, we compared our method with the latest clustering-based method CC-CVRS (Constrained Clustering Capacitated Vehicle Routing Solvers, 2022) [1]. The CC-CVRS uses a constrained clustering approach to decompose the large-scale CVRPs into several small-scale clusters, each vehicle could visit one or several clusters. They also focused on large-scale  CVRP.  We compared TAM with CC-CVRS on all instances with over 500 nodes in Uchoa et al. (2014) of CVRPLIB. The gap and solving time are reported in the following tables. The results show that our TAM-LKH3 trained on CVRP 100 could obtain competitive results as the CC-CVRS while consuming much less time.
>
> The gap of instances
> | Length      | No. Nodes | AM     | TAM-AM | LKH3   | TAM-LKH3   | CC-CVRS |
> |-------------|-----------|--------|--------|--------|------------|---------|
> | X-n502-k39  | 502       | 13.97% | 6.45%  | 1.02%  | 5.75%      | 2.50%   |
> | X-n513-k21  | 513       | 18.13% | 11.47% | 3.50%  | **9.41%**  | 17.40%  |
> | X-n524-k153 | 524       | 27.57% | 16.73% | 3.91%  | 14.34%     | -       |
> | X-n536-k96  | 536       | 27.67% | 10.32% | 19.05% | 10.25%     | -       |
> | X-n548-k50  | 548       | 28.09% | 14.30% | 5.38%  | 13.71%     | 5.00%   |
> | X-n561-k42  | 561       | 17.85% | 10.60% | 8.33%  | **9.36%**  | 13.10%  |
> | X-n573-k30  | 573       | 25.79% | 10.50% | 4.77%  | 7.56%      | 6.60%   |
> | X-n586-k159 | 586       | 19.40% | 8.90%  | 12.73% | 8.89%      | 5.10%   |
> | X-n599-k92  | 599       | 24.02% | 9.56%  | 34.68% | 9.37%      | 5.10%   |
> | X-n613-k62  | 613       | 18.20% | 11.64% | 16.38% | **9.69%**  | 13.60%  |
> | X-n627-k43  | 627       | 51.64% | 10.31% | 17.23% | 9.61%      | 6.90%   |
> | X-n641-k35  | 641       | 24.62% | 8.56%  | 7.93%  | **7.39%**  | 9.70%   |
> | X-n655-k131 | 655       | 20.55% | 7.31%  | 3.03%  | 6.66%      | 1.00%   |
> | X-n670-k130 | 670       | 31.18% | 19.29% | 18.55% | 19.01%     | -       |
> | X-n685-k75  | 685       | 19.80% | 11.03% | 17.59% | **10.40%** | 16.90%  |
> | X-n701-k44  | 701       | 29.56% | 7.73%  | 8.87%  | **6.81%**  | 9.80%   |
> | X-n716-k35  | 716       | 32.90% | 12.00% | 9.35%  | 10.05%     | 8.60%   |
> | X-n733-k159 | 733       | 18.74% | 8.60%  | 16.65% | 8.51%      | 8.20%   |
> | X-n749-k98  | 749       | 17.06% | 10.22% | 19.90% | 10.03%     | 7.90%   |
> | X-n766-k71  | 766       | 29.50% | 9.84%  | 12.44% | **8.82%**  | 11.70%  |
> | X-n783-k48  | 783       | 15.60% | 12.95% | 11.87% | **10.23%** | 11.30%  |
> | X-n801-k40  | 801       | 34.30% | 11.91% | 5.57%  | 10.34%     | 8.50%   |
> | X-n819-k171 | 819       | 16.02% | 9.70%  | 28.92% | 9.60%      | -       |
> | X-n837-k142 | 837       | 14.50% | 7.58%  | 11.16% | 7.51%      | -       |
> | X-n856-k95  | 856       | 12.27% | 8.99%  | 8.09%  | 8.68%      | -       |
> | X-n876-k59  | 876       | 33.15% | 9.37%  | 9.46%  | 8.10%      | 5.80%   |
> | X-n957-k87  | 957       | 36.06% | 16.94% | 10.43% | 16.53%     | 6.10%   |
> | X-n895-k37  | 895       | 36.84% | 14.32% | 29.72% | **9.39%**  | 12.20%  |
> | X-n916-k207 | 916       | 17.19% | 8.91%  | 10.34% | 8.87%      | -       |
> | X-n936-k151 | 936       | 27.86% | 13.92% | 22.89% | 13.47%     | -       |
> | X-n979-k58  | 979       | 30.41% | 8.23%  | 15.80% | 6.70%      | 5.30%   |
> | X-n1001-k43 | 1001      | 18.36% | 12.99% | 13.79% | **10.64%** | 13.60%  |
>
> [1] Alesiani F, Ermis G, Gkiotsalitis K. Constrained Clustering for the Capacitated Vehicle Routing Problem (CC-CVRP)[J]. Applied artificial intelligence, 2022, 36(1): 1995658.

---

> > ### Author Response · Authors · 2022-12-08
> > **Response to Reviewer 7d5S-Solving time of instances**
> >
> > | Time        | No. Nodes | AM    | TAM-AM | LKH3  | TAM-LKH3 | CC-CVRS |
> > |-------------|-----------|-------|--------|-------|----------|---------|
> > | X-n502-k39  | 502       | 0.74  | 1.01   | 2.05  | 1.47     | 100.00  |
> > | X-n513-k21  | 513       | 0.68  | 0.99   | 2.10  | 1.41     | 100.00  |
> > | X-n524-k153 | 524       | 0.88  | 1.00   | 2.78  | 1.30     | -       |
> > | X-n536-k96  | 536       | 0.83  | 0.95   | 2.37  | 1.28     | -       |
> > | X-n548-k50  | 548       | 0.78  | 0.88   | 2.05  | 1.18     | 100.00  |
> > | X-n561-k42  | 561       | 0.80  | 0.93   | 2.08  | 1.31     | 100.00  |
> > | X-n573-k30  | 573       | 0.79  | 0.99   | 2.05  | 1.97     | 100.00  |
> > | X-n586-k159 | 586       | 1.00  | 1.08   | 3.59  | 1.58     | 100.00  |
> > | X-n599-k92  | 599       | 1.00  | 1.13   | 2.33  | 1.52     | 100.00  |
> > | X-n613-k62  | 613       | 0.92  | 1.02   | 2.34  | 1.39     | 100.00  |
> > | X-n627-k43  | 627       | 0.98  | 1.01   | 2.15  | 1.71     | 100.00  |
> > | X-n641-k35  | 641       | 1.03  | 1.53   | 2.10  | 2.03     | 100.00  |
> > | X-n655-k131 | 655       | 1.07  | 1.13   | 3.63  | 1.63     | 100.00  |
> > | X-n670-k130 | 670       | 1.09  | 1.24   | 4.97  | 1.64     | -       |
> > | X-n685-k75  | 685       | 1.01  | 1.16   | 2.62  | 1.72     | 100.00  |
> > | X-n701-k44  | 701       | 1.03  | 1.16   | 3.04  | 1.51     | 100.00  |
> > | X-n716-k35  | 716       | 1.05  | 1.23   | 3.06  | 1.92     | 100.00  |
> > | X-n733-k159 | 733       | 1.25  | 1.36   | 4.82  | 1.89     | 100.00  |
> > | X-n749-k98  | 749       | 1.61  | 1.55   | 3.42  | 1.95     | 100.00  |
> > | X-n766-k71  | 766       | 1.16  | 1.34   | 4.28  | 2.01     | 100.00  |
> > | X-n783-k48  | 783       | 1.14  | 1.35   | 4.06  | 2.74     | 100.00  |
> > | X-n801-k40  | 801       | 1.46  | 1.63   | 4.12  | 2.47     | 100.00  |
> > | X-n819-k171 | 819       | 1.50  | 1.81   | 4.65  | 2.44     | -       |
> > | X-n837-k142 | 837       | 1.37  | 1.49   | 5.34  | 2.06     | -       |
> > | X-n856-k95  | 856       | 1.30  | 1.40   | 5.27  | 2.10     | -       |
> > | X-n876-k59  | 876       | 1.29  | 1.43   | 5.20  | 2.66     | 100.00  |
> > | X-n957-k87  | 957       | 3.42  | 2.91   | 4.21  | 5.01     | 100.00  |
> > | X-n895-k37  | 895       | 2.64  | 2.58   | 4.23  | 3.91     | 100.00  |
> > | X-n916-k207 | 916       | 3.16  | 3.09   | 8.93  | 6.84     | -       |
> > | X-n936-k151 | 936       | 3.94  | 3.41   | 9.22  | 6.05     | -       |
> > | X-n979-k58  | 979       | 3.03  | 2.95   | 4.22  | 5.06     | 100.00  |
> > | X-n1001-k43 | 1001      | 2.95  | 2.93   | 4.18  | 4.68     | 100.00  |

---

### Official Review · Reviewer_o9TS · 2022-10-25

**Confidence:** 4
**Correctness:** 4
**Technical Novelty And Significance:** 3
**Empirical Novelty And Significance:** 2
**Recommendation:** 8

**Clarity, Quality, Novelty And Reproducibility:**

This paper is well motivated. The experiments are relatively comprehensive and convincing. The idea is simple but effective, which has practical values. The code is not provided though. As the implementation seems to be nontrivial, the reproducibility might be an issue.


**Strength And Weaknesses:**

Strength:

- The problem tackled by the paper is an important combinatorial optimization that is actively used in daily work.
- The proposed divide-and-conquer paradigm is reasonable and effective.
- The empirical results show that the paper is able to achieve better results compared to some existing solvers. Also the paper is able to incorporate existing TSP solvers as subroutines and combine the power of both learning and existing human designed heuristics.

Weakness:

- The paper tackles a limited version of the CVRP. In practice the problem might be more complicated that it is hard to first divide the nodes while still obtaining feasible solutions. For example, in practice there would be constraints like 1) a driver could not drive more than certain hours; 2) certain packages should be delivered to a certain location within a given time period.

- Compared to LKH3 or ORTOOLS the proposed approach obtained comparable or better results with a shorter amount of time. However in practice it is not clear if the proposed approach would further improve with more time. In other words it would be interesting to see the time-solution quality trade-offs of this method and other iteration-based solvers.


**Summary Of The Paper:**

This paper presents a framework for learning to solve VRP. The core idea is to first learn a policy that can divide the routes into sub-routes where each sub-route is covered by a single vehicle. The one can use efficient TSP solvers for each of the sub-route, and also solve them in parallel. Experiments show that the proposed approach is able to learn the solving from small-scaled datasets, and generalize the results to large graph with more than 5000 nodes.


**Summary Of The Review:**

Overall a practical paper for VRP with two-stage approaches. It is a nice combination of learning and existing heuristics. The paper can potentially be improved with more analysis of the results, and directions on handling practical constraints when deploying VRP to real-world problems.

---

> ### Author Response · Authors · 2022-11-18
> **Response to Reviewer o9TS**
>
> We are extremely grateful for your recognition and valuable comments.  We agree with you that the idea is not complex but effective, and has practical values. In fact, our TAM and its variants with time window have been used for real-world applications in real-time due to their superior performance and computation efficiency.  We have carefully thought about your comments. In the following part, we will try our best to address your main concerns.
>
> **Considering more practical constraints in TAM**
>
> We agree with you that this paper focuses on CVRP with maximum vehicle number constraints.  It's also true that it is harder to divide the node while satisfying complex constraints such as time window and capacity constraints.  However, our TAM could take the complex constraints into consideration when dividing the node. Due to the proposed new reward and seq2seq reformulation, TAM can easily encode practical constraints by changing the mask function without modifying dividing models.
>
> For instance, the time window constraints could be considered in TAM with the following mask function.
> If the current vehicle cannot arrive the node $i$ in the time window $[T^l_i, T^u_i ]$, then the current vehicle cannot visit node $i$. Please refer to Appendix A.3.4 for the full mask function of the time window constraint.
>
> **The trade-off between the time and solution quality**
>
> We agree with you that the trade-off between time and solution quality is interesting.  For our TAM, the trade-off could be achieved by changing beam size or sample size of search stragety.  We have compared three search strategies (greedy search, beam search, and sample search) and discussed the trade-off in Appendix A.3.2. For example, sample strategy could sample totally different splits and could greatly improve the results of our TAM with a large sample number. However, the computation time and memory are also increasing with the sample number. For instance, Figure 16 shows that TAM-LKH3 with sample size 1000 could find the best solution 6.84 among all methods for a real-world CVRP 388 instance, while TAM-LKH3 with beam size 10 finds a solution 6.99. But the solving time also increases from 3.73s to 17.52s.
>
> **Reproducibility**
>
> As you pointed out, our TAM with the proposed three generalization techniques (generating subroute sequence, new reward, and global mask function) is not complex but effective. It can be easily implemented based on seq2seq model (such as Attention Model and GNN) and open-sourced solver (such as LKH3 and Ortoools).  The details of the training algorithm implementation are shown in  Algorithm 1 in Appendix A.3.1. There are three implementation differences between TAM and traditional RL method: 1) training an Attention Model for solving TSP in advance, 2) padding all sub-TSPs to the same length, 3) calculate the reward with Equation 18.  As suggested, we add the implementation details in Algorithm 2 and Algorithm 3 in Appendix of the revised version for the inference algorithm to improve reproducibility. If the paper is accepted. we will also release an implementation once getting necessary permission.

---

> > ### Comment · Reviewer_o9TS · 2022-11-26
> > **thanks!**
> >
> > Thanks for your reply! I think this paper is a nice contribution to the field. Look forward to your code release.

---

> > > ### Author Response · Authors · 2022-12-08
> > > **Thanks for reviewing！**
> > >
> > > Thanks for taking the time to review our paper carefully and provide valuable suggestions.

---

### Official Review · Reviewer_cdJ3 · 2022-11-04

**Confidence:** 5
**Correctness:** 3
**Technical Novelty And Significance:** 3
**Empirical Novelty And Significance:** 3
**Recommendation:** 6

**Clarity, Quality, Novelty And Reproducibility:**

The paper is mostly clear and well written.
The algorithm is described well and should be reproducible.
The idea of the paper is not novelty and it is explored before. But, the overall framework seems working well, which was not the case in previous papers.

**Strength And Weaknesses:**

Strength
A new RL algorithm is proposed to solve large-scale VRP problems.

Weakness
The numerical experiments and benchmarks could be improved.

**Summary Of The Paper:**

An RL algorithm, TAM, is proposed to solve large-scale VRP problems.  TAM learns how to split the problem into smaller problems and then solve the smaller problems which are instances of TSP by well-performed heuristic algorithms like LKH3. The splitting problem is defined as an RL problem in which the state of the system is the current partial solution which is obtained by the chosen sub-routes of the VRP problem. The action is choosing an unvisited sub-route which is the set of customers/nodes that a vehicle could serve in one round of service. The reward is the minimal distance of the chosen sub-route. To get that, for a given sub-rout, the minimum distance is obtained by using an optimal algorithm or any fast heuristic (here a learned model by RL is used to exploit the power of parallel computation by GPUs).
The goal is to minimize the total travel length.
To assign the nodes to the sub-routes, the same algorithm as the AM model, proposed by Kool et. al (2019) is used where the masking operator is added to the decoder to assign the nodes into the sub-routes. To use the GPU for solving the TSP problem in the sub-routes, the padding mechanism is used. For each sub-route problem, the trained RL models of sizes 20, 50, and 100 are available and the model which is closest to the size of each sub-route problem is used.

I really enjoyed reading the paper and the results are impressive to some extent.


**Summary Of The Review:**

I only have two questions from the authors which upon getting satisfying answer, I'll raise my vote.

Q1- I did not get "If the capacity of unused vehicles cannot serve the rest demand, then the current vehicle can not visit the depot". I am not sure why the capacity of the rest of the network is considered. If you are choosing a given set of nodes to assign them to a sub-route, then, you can easily check the sum of demand of those nodes. Once the sum of demands of those nodes gets bigger than the capacity of the vehicle, you stop assigning the nodes to the vehicle. Given this, why do you need the new masking operator? This procedure should end with the same result as the suggested masking operator, though it is cheaper and more intuitive.

Q2- In the presented results in Table 1, I do not see the optimal solution. For some of the presented instances, you can get the optimal solution with the branch & price algorithm presented in [1]. Besides, the main comment on the numerical experiment is on the benchmark set. If you want to show the true value of your algorithm you need to demonstrate the performance of your algorithm on the CVRPLIB problem set (http://vrp.atd-lab.inf.puc-rio.br/index.php/en/) and instance in [3, 4]. The ultimate benchmarking would involve comparing your results with the state-of-the-art algorithm, like [2, 4, 5], which has some of the best results on that problem set.


[1] Pessoa, Artur, Ruslan Sadykov, Eduardo Uchoa, and François Vanderbeck. "A generic exact solver for vehicle routing and related problems." Mathematical Programming 183, no. 1 (2020): 483-523.

[2] Vidal, Thibaut. "Hybrid genetic search for the CVRP: Open-source implementation and SWAP* neighborhood." Computers & Operations Research 140 (2022): 105643.

[3] Uchoa, E., et al. (2017). New benchmark instances for the capacitated vehicle routing problem. European Journal of Operational Research, 257(3), 845–858.

[4] Arnold, F., et al. (2019). Efficiently solving very large scale routing problems. Computers & Operations Research, 107(1), 32–42.

[5] Christiaens, J., & Vanden Berghe, G. (2020). Slack induction by string removals for vehicle routing problems. Transportation Science, 54(2), 299–564.

---

> ### Author Response · Authors · 2022-11-18
> **Response to Reviewer cdJ3-Part I**
>
> We really appreciate your recognition and are glad to hear that you enjoyed reading our paper.  Your summary of our paper is very impressive.  We also agree with you that the idea of Divider and Conquer is widely used and not novel. But learning the best global split by reinforcement learning while considering complex constraints such as capacity, time window, and maximum vehicle number is still challenging. In addition, we mainly contribute to generalizing the learned heuristics trained on small-scale VRPs to solve large-scale VRPs in a real-time and zero-shot way by generating sub-route sequence rather than node sequence. To the best of our knowledge, our TAM with the three proposed generalization techniques is also the first learn-to-construct method that can successfully scale to solve VRPs with over 5000 nodes in seconds.  As suggested, we will try our best to address your two main questions in the following part.
>
> > **Q1-I did not get "If the capacity of unused vehicles cannot serve the rest demand, then the current vehicle can not visit the depot". I am not sure why the capacity of the rest of the network is considered.**
>
> You correctly point out that the capacity of a sub-route could be easily checked by the sum of the demand of its nodes. It's also the core idea behind the local mask function for each vehicle's capacity constraints in A.3.4 in Appendix. If the rest capacity of the current vehicle ($C-c$) cannot serve the demand of node $i$ ($d_i$), then the current vehicle cannot visit node $i$.
>
> Our new global mask function is designed for global maximum vehicle number constraints.  The rest of the demand and capacity of unused vehicles are considered in the mask function to prevent the potential violation of maximum vehicle number constraints while keeping the optimal action space.  It's true that the rule, "Once the sum of demands of those nodes gets bigger than the capacity of the vehicle, you stop assigning the nodes to the vehicle", could help us prevent the potential violation of maximum vehicle number constraints. But it would make each vehicle carry as much demand as possible, which would reduce our action space. Our dividing model is trained by a reward with minimum travel length rather than making each vehicle carry as much demand as possible, it is then highly possible that the current vehicle that does not carry more demand would be the optimal solution even if the capacity is not used up. Our new mask function could always keep the probability to choose the above actions while satisfying the global maximum vehicle number constraints.  For the above consideration, we proposed a new global mask function.
>
> > **Q2-More benchmark set and optimal solution.**
>
> We have thought very carefully about your comment regarding the optimal solutions benchmark set.  As you pointed out, the optimal solutions of some instances could be obtained by exact methods such as branch and price. It's true that the optimal solution is an important reference for our method. But this paper's main topic is generalizing the learned heuristics to obtain acceptable large-scale VRPs solutions for real-time applications. Obtaining optimal solutions for large-scale VRPs in seconds is intractable and time-consuming. In addition, we want to test the stability and average performance of our TAM on 100 large-scale instances, which makes obtaining optimal solutions more difficult.  That's why we turn to choose the same benchmark set and method in AM and POMO for the original manuscripts. The experiment results show our TAM could outperform AM, POMO, DPDP, Ortools, and LKH3 on large-scale VRPs with over 1000 nodes in seconds.  Our TAM has been used for real-world and real-time applications, we also show and visualize the TAM's performance on real-world instances in Table 2 and Figures 10-16.
>
> As you suggested, we have tried our best to compare our TAM with more benchmark methods from OR domain. We have tested our TAM and the open-sourced HGS on 100 instances of CVRP 5000 and 7000 in real-time. The following Tables report the average route length and solving time of the instances. HGS-t60 means the HGS with 60s time limit. We also implement the TAM-HGS, which uses HGS as the second stage solver as TAM. For real-time application, we found the HGS cannot find a solution in 60 seconds while TAM-HGS could find a good solution in 30.23s and 52.36s for CVRP 5000 and CVRP 7000 respectively. It's also interesting to see that TAM-HGS consumes more time than TAM-LKH3 while finding the nearly same quality solutions.
>
> | Length    | AM     | TAM-AM | LKH3   | TAM-LKH3 | TAM-HGS | HGS-t60 |
> |-----------|--------|--------|--------|----------|---------|---------|
> | CVRP-5000 | 257.06 | 172.22 | 175.66 | 144.64   | 142.83  | -       |
> | CVRP-7000 | 354.28 | 233.44 | 244.97 | 196.91   | 193.64  | -       |

---

> > ### Author Response · Authors · 2022-11-18
> > **Response to Reviewer cdJ3-Part II**
> >
> > Time of synthetic instances
> > | Time      | AM     | TAM-AM | LKH3    | TAM-LKH3 | TAM-HGS | HGS-t60 |
> > |-----------|--------|--------|---------|----------|---------|---------|
> > | CVRP-5000 | 11.50  | 11.78  | 151.64  | 17.19    | 30.23   | 60.00   |
> > | CVRP-7000 | 26.13  | 26.47  | 501.26  | 33.21    | 52.36   | 60.00   |
> >
> > Following your suggestions, we also test our TAM on several very-large instances on CVRPLIB.  The following Tables report the gap and solving time of the instances.  The results show that our TAM-LKH3 could find significantly better results than AM and could find a better solution much faster than LKH3.  We also observed that the gap between our TAM and the best solution is about 5-10\% when generalizing to VRP with about 1000 nodes, and about 15-25\% when generalizing to VRP with over 3000-10000 nodes.  This is acceptable for real-time applications given that we limit the maximum solving time of our TAM as 60 seconds and the beam search size as 10 for real-time applications.  There are two main reasons behind the gap difference: 1) larger-scale VRP is much harder to solve; 2) the difference between large-scale VRP and VRP 100 (used for training TAM) enlarges as the nodes number increases.   If the optimal solutions are required for non-real-time applications, we should significantly improve the beam size or sample size of our TAM to find a better solution.
> >
> > Gap of CVRPLIB instances
> > | Length        | No. Nodes | AM     | TAM-AM | LKH3   | TAM-LKH3 |
> > |---------------|-----------|--------|--------|--------|----------|
> > | X-n895-k37    | 895       | 36.84% | 14.32% | 29.72% | 9.39%    |
> > | X-n916-k207   | 916       | 17.19% | 8.91%  | 10.34% | 8.87%    |
> > | X-n936-k151   | 936       | 27.86% | 13.92% | 22.89% | 13.47%   |
> > | X-n979-k58    | 979       | 30.41% | 8.23%  | 15.80% | 6.70%    |
> > | X-n1001-k43   | 1001      | 18.36% | 12.99% | 13.79% | 10.64%   |
> > | Leuven1-3001  | 3001      | 46.93% | 20.24% | 18.10% | 19.30%   |
> > | Leuven2-4001  | 4001      | 53.31% | 38.57% | 22.14% | 15.88%   |
> > | Antwerp1-6001 | 6001      | 39.30% | 24.90% | 24.20% | 24.01%   |
> > | Antwerp2-7001 | 7001      | 50.32% | 33.20% | 31.09% | 22.55%   |
> > | Ghent1-10001  | 10001     | 46.89% | 30.20% | -      | 29.53%   |
> > | Ghent2-11001  | 11001     | 52.20% | 33.29% | -      | 23.65%   |
> >
> > Time of CVRPLIB instances
> > | Time               | No. Nodes | AM     | TAM-AM | LKH3    | TAM-LKH3 |
> > |--------------------|-----------|--------|--------|---------|----------|
> > |        X-n895-k37  | 895       | 2.64   | 2.58   | 4.23    | 3.91     |
> > |        X-n916-k207 | 916       | 3.16   | 3.09   | 8.93    | 6.84     |
> > | X-n936-k151        | 936       | 3.94   | 3.41   | 9.22    | 6.05     |
> > | X-n979-k58         | 979       | 3.03   | 2.95   | 4.22    | 5.06     |
> > | X-n1001-k43        | 1001      | 2.95   | 2.93   | 4.18    | 4.68     |
> > | Leuven1-3001       | 3001      | 9.73   | 9.82   | 68.99   | 15.59    |
> > | Leuven2-4001       | 4001      | 13.45  | 13.68  | 73.96   | 23.73    |
> > | Antwerp1-6001      | 6001      | 12.81  | 12.98  | 596.21  | 24.91    |
> > | Antwerp2-7001      | 7001      | 14.96  | 15.26  | 479.47  | 31.90    |
> > | Ghent1-10001       | 10001     | 21.41  | 21.66  | -       | 37.26    |
> > | Ghent2-11001       | 11001     | 38.55  | 37.56  | -       | 55.73    |

---

> > > ### Comment · Reviewer_cdJ3 · 2022-11-30
> > > **CVRPLIB instances**
> > >
> > > Thanks for running those experiments on CVRPLIB problems.
> > > I was wondering how did you choose these instances to report? Since you do not need to re-train your model for the new dataset and running the inference is quite fast, I would suggest reporting the results of all CVRPLIB instances that have more nodes than X (like X=500). Otherwise, people may think that these results are kind of cherry-picked and it is not possible to get solutions of similar quality for other problems.

---

> > > > ### Author Response · Authors · 2022-11-30
> > > > **Response to CVRPLIB instances**
> > > >
> > > > Thanks for your valuable suggestions.  The instances from CVRPLIB are not cherry-picked. We choose the instances from two widely used CVRPLIB datasets **Uchoa et al. (2014)** and **Arnold, Gendreau and Sörensen (2017)**[1].  **The X is chosen as 900**. All instances in the two datasets with more than 900 nodes are chosen and reported in our results. The instance X-n895-k37 is reported because it has 895 nodes, which is very close to 900. The instances with over 12000 nodes such as Brussels1 and Brussels2 are not reported because the attention matrix of attention model will consume a lot of memory. On a single GPU, our TAM with greedy search cannot solve them within 60 seconds due to out of memory of the GPU.  If we extend the time limit to 180 seconds, Brussels1 and Brussels2 could be solved with the greedy search on CPU. The results are shown in the following tables.  The results show that our TAM can still significantly reduce the gap in a reasonable time.  As we generalize our TAM learned on VRP 100 to solve very large instances with over 12000 nodes, both the gap and solving time will also increase.
> > > >
> > > > It should be noted that if we want to solve very large instances with over 12000 nodes faster and obtain better solutions, we should run TAM with beam search or sample search on GPU with larger memory.
> > > >
> > > > Gap of Brussels1 and Brussels2 instances
> > > > | Length          | No. Nodes | AM     | TAM-AM | LKH3 | TAM-LKH3 |
> > > > |-----------------|-----------|--------|--------|------|----------|
> > > > | Brussels1-15001 | 15001     | 52.37% | 43.42% | -    | 27.15%   |
> > > > | Brussels2-16001 | 16001     | 52.44% | 39.04% | -    | 37.07%   |
> > > >
> > > > Time of Brussels1 and Brussels2 instances
> > > > | Time            | No. Nodes | AM      | TAM-AM  | LKH3 | TAM-LKH3 |
> > > > |-----------------|-----------|---------|---------|------|----------|
> > > > | Brussels1-15001 | 15001     | 131.39  | 139.09  | -    | 166.81   |
> > > > | Brussels2-16001 | 16001     | 165.94  | 158.71  | -    | 187.48   |
> > > >
> > > > [1] http://vrp.galgos.inf.puc-rio.br/index.php/en/

---

> > > > > ### Comment · Reviewer_cdJ3 · 2022-12-04
> > > > > **Response**
> > > > >
> > > > > I think it would be beneficial if at first you mentioned that you run only the instances of those two papers. Anyways, among the Uchoa et al. (2014) instances with more than 900 nodes, I think one instance is missing in your report, "X-n957-k87". Would you be able to report the cost of that one as well?
> > > > > Besides, if you wanted to report the results of only big instances, not all the instances in VRPLIB, I still do not see any reason to not report the results of instance in "Li et al. (2005)". Given that you only need to run a forward-pass on your trained network, it should be very easy and quick to get the answer on these instances too. I am asking that since on OR community it is important to know all the strength and weakness of an algorithm compared to the previous competitors which only can be known if you would be able to run benchmarking on all existing benchmarks.
> > > > > Because of that, still, I would suggest you, to increase the quality of numerical experiments, to provide the benchmarking results  over all medium and large size instances, i.e., instance with around 200 nodes and bigger. This will create a solid benchmark for the future researches.

---

> > > > > > ### Author Response · Authors · 2022-12-04
> > > > > > **Response to CVRPLIB instances-Part II**
> > > > > >
> > > > > > We are really grateful for taking your time to review our results carefully and provide valuable suggestions, which help improve our work a lot. We have carefully thought about your comments. In the following part, we will try our best to follow your suggestions.
> > > > > >
> > > > > > As pointed out, we have added the instance "X-n957-k87" to our benchmark.  We also agree with you that more results will make a more solid benchmark, especially for OR community.  We choose the original X as 900 because the scope of the paper is focused on generalizing to solve large-scale VRPs in real-time.  Following your suggestions, we have reported the results of all **52** medium and large instances with over 500 nodes in the following tables to enhance our benchmark. Considering the current long list here, we will report the results of instances between 200 and 500 nodes in the Appendix of the final version of the paper (there are more than 130 instances with over 200 nodes). It should be noted that the following cases with over 500 nodes are not included in the 52 instances and are not reported:
> > > > > > 1) the results of Flanders1 and Flanders2 with over 20000 nodes are not reported because they cannot be solved in real-time.
> > > > > > 2) the results of DIMACS (2021) are not reported because they use an explicit distance matrix while this paper focuses on euclidean distance.
> > > > > >
> > > > > > Following your suggestions, we have also added the results of "Li et al." in the following tables.  We do not report the results of instances of "Li et al. (2005)" in the previous version because we found that the instances are specially designed [1], which may be not suitable for real-world applications. In addition, the instances of "Li et al. (2005)" are not standard CVRP [2].  It should be noted that the reported results of Li et al. (2005) could be better than the current best solution because the instances in Li et al. (2005) are transformed into the standard CVRP by removing length constraints.
> > > > > >
> > > > > > As shown in the tables, TAM is significantly better than AM with similar solving time. On average, TAM could obtain better solutions than LKH-3 while consuming much less time. For very large instances with over 5000 nodes, the gap between TAM and LKH-3 becomes larger.
> > > > > >
> > > > > > [1] http://vrp.galgos.inf.puc-rio.br/index.php/en/plotted-instances?data=Li_21
> > > > > >
> > > > > > [2] Li F, Golden B, Wasil E. Very large-scale vehicle routing: new test problems, algorithms, and results[J]. Computers & Operations Research, 2005, 32(5): 1165-1179.

---

> > > > > > > ### Author Response · Authors · 2022-12-04
> > > > > > > **The gap of CVRPLIB instances**
> > > > > > >
> > > > > > > | Length          | No. Nodes | AM         | TAM-AM     | LKH3       | TAM-LKH3   |
> > > > > > > |-----------------|-----------|------------|------------|------------|------------|
> > > > > > > | X-n502-k39      | 502       | 13.97%     | 6.45%      | 1.02%      | 5.75%      |
> > > > > > > | X-n513-k21      | 513       | 18.13%     | 11.47%     | 3.50%      | 9.41%      |
> > > > > > > | X-n524-k153     | 524       | 27.57%     | 16.73%     | 3.91%      | 14.34%     |
> > > > > > > | X-n536-k96      | 536       | 27.67%     | 10.32%     | 19.05%     | 10.25%     |
> > > > > > > | X-n548-k50      | 548       | 28.09%     | 14.30%     | 5.38%      | 13.71%     |
> > > > > > > | X-n561-k42      | 561       | 17.85%     | 10.60%     | 8.33%      | 9.36%      |
> > > > > > > | X-n573-k30      | 573       | 25.79%     | 10.50%     | 4.77%      | 7.56%      |
> > > > > > > | X-n586-k159     | 586       | 19.40%     | 8.90%      | 12.73%     | 8.89%      |
> > > > > > > | X-n599-k92      | 599       | 24.02%     | 9.56%      | 34.68%     | 9.37%      |
> > > > > > > | X-n613-k62      | 613       | 18.20%     | 11.64%     | 16.38%     | 9.69%      |
> > > > > > > | X-n627-k43      | 627       | 51.64%     | 10.31%     | 17.23%     | 9.61%      |
> > > > > > > | X-n641-k35      | 641       | 24.62%     | 8.56%      | 7.93%      | 7.39%      |
> > > > > > > | X-n655-k131     | 655       | 20.55%     | 7.31%      | 3.03%      | 6.66%      |
> > > > > > > | X-n670-k130     | 670       | 31.18%     | 19.29%     | 18.55%     | 19.01%     |
> > > > > > > | X-n685-k75      | 685       | 19.80%     | 11.03%     | 17.59%     | 10.40%     |
> > > > > > > | X-n701-k44      | 701       | 29.56%     | 7.73%      | 8.87%      | 6.81%      |
> > > > > > > | X-n716-k35      | 716       | 32.90%     | 12.00%     | 9.35%      | 10.05%     |
> > > > > > > | X-n733-k159     | 733       | 18.74%     | 8.60%      | 16.65%     | 8.51%      |
> > > > > > > | X-n749-k98      | 749       | 17.06%     | 10.22%     | 19.90%     | 10.03%     |
> > > > > > > | X-n766-k71      | 766       | 29.50%     | 9.84%      | 12.44%     | 8.82%      |
> > > > > > > | X-n783-k48      | 783       | 15.60%     | 12.95%     | 11.87%     | 10.23%     |
> > > > > > > | X-n801-k40      | 801       | 34.30%     | 11.91%     | 5.57%      | 10.34%     |
> > > > > > > | X-n819-k171     | 819       | 16.02%     | 9.70%      | 28.92%     | 9.60%      |
> > > > > > > | X-n837-k142     | 837       | 14.50%     | 7.58%      | 11.16%     | 7.51%      |
> > > > > > > | X-n856-k95      | 856       | 12.27%     | 8.99%      | 8.09%      | 8.68%      |
> > > > > > > | X-n876-k59      | 876       | 33.15%     | 9.37%      | 9.46%      | 8.10%      |
> > > > > > > | X-n957-k87      | 957       | 36.06%     | 16.94%     | 10.43%     | 16.53%     |
> > > > > > > | X-n895-k37      | 895       | 36.84%     | 14.32%     | 29.72%     | 9.39%      |
> > > > > > > | X-n916-k207     | 916       | 17.19%     | 8.91%      | 10.34%     | 8.87%      |
> > > > > > > | X-n936-k151     | 936       | 27.86%     | 13.92%     | 22.89%     | 13.47%     |
> > > > > > > | X-n979-k58      | 979       | 30.41%     | 8.23%      | 15.80%     | 6.70%      |
> > > > > > > | X-n1001-k43     | 1001      | 18.36%     | 12.99%     | 13.79%     | 10.64%     |
> > > > > > > |        Li_21    | 561       | 31.91%     | 8.87%      | 0.95%      | 4.02%      |
> > > > > > > |        Li_22    | 601       | 24.03%     | 4.73%      | -2.50%     | -0.27%     |
> > > > > > > |        Li_23    | 641       | 35.43%     | 13.38%     | 1.34%      | 6.34%      |
> > > > > > > |        Li_24    | 721       | 43.14%     | 14.82%     | 2.86%      | 7.15%      |
> > > > > > > |        Li_25    | 761       | 31.28%     | 7.38%      | -2.76%     | -1.56%     |
> > > > > > > |        Li_26    | 801       | 42.38%     | 18.77%     | 1.52%      | 10.80%     |
> > > > > > > |        Li_27    | 841       | 31.74%     | 9.00%      | -4.14%     | 2.35%      |
> > > > > > > |        Li_28    | 881       | 50.23%     | 16.84%     | 1.73%      | 10.79%     |
> > > > > > > |        Li_29    | 961       | 47.67%     | 19.34%     | 1.96%      | 10.26%     |
> > > > > > > |        Li_30    | 1041      | 52.34%     | 24.89%     | 4.02%      | 12.29%     |
> > > > > > > |        Li_31    | 1121      | 52.77%     | 31.45%     | 2.55%      | 13.41%     |
> > > > > > > |        Li_32    | 1201      | 53.43%     | 30.96%     | 1.62%      | 14.69%     |
> > > > > > > | Leuven1-3001    | 3001      | 46.93%     | 20.24%     | 18.10%     | 19.30%     |
> > > > > > > | Leuven2-4001    | 4001      | 53.31%     | 38.57%     | 22.14%     | 15.88%     |
> > > > > > > | Antwerp1-6001   | 6001      | 39.30%     | 24.90%     | 24.20%     | 24.01%     |
> > > > > > > | Antwerp2-7001   | 7001      | 50.32%     | 33.20%     | 31.09%     | 22.55%     |
> > > > > > > | **Average**     | **1110** | **30.73%** | **13.93%** | **10.92%** | **10.16%** |
> > > > > > > | Ghent1-10001    | 10001     | 46.89%     | 30.20%     | -          | 29.53%     |
> > > > > > > | Ghent2-11001    | 11001     | 52.20%     | 33.29%     | -          | 23.65%     |
> > > > > > > | Brussels1-15001 | 15001     | 52.37%     | 43.42%     | -          | 27.15%     |
> > > > > > > | Brussels2-16001 | 16001     | 52.44%     | 39.04%     | -          | 37.07%     |

---

> > > > > > > > ### Author Response · Authors · 2022-12-04
> > > > > > > > **The time of CVRPLIB instances**
> > > > > > > >
> > > > > > > > | Time            | No. Nodes | AM        | TAM-AM    | LKH3       | TAM-LKH3  |
> > > > > > > > |-----------------|-----------|-----------|-----------|------------|-----------|
> > > > > > > > | X-n502-k39      | 502       | 0.74      | 1.01      | 2.05       | 1.47      |
> > > > > > > > | X-n513-k21      | 513       | 0.68      | 0.99      | 2.10       | 1.41      |
> > > > > > > > | X-n524-k153     | 524       | 0.88      | 1.00      | 2.78       | 1.30      |
> > > > > > > > | X-n536-k96      | 536       | 0.83      | 0.95      | 2.37       | 1.28      |
> > > > > > > > | X-n548-k50      | 548       | 0.78      | 0.88      | 2.05       | 1.18      |
> > > > > > > > | X-n561-k42      | 561       | 0.80      | 0.93      | 2.08       | 1.31      |
> > > > > > > > | X-n573-k30      | 573       | 0.79      | 0.99      | 2.05       | 1.97      |
> > > > > > > > | X-n586-k159     | 586       | 1.00      | 1.08      | 3.59       | 1.58      |
> > > > > > > > | X-n599-k92      | 599       | 1.00      | 1.13      | 2.33       | 1.52      |
> > > > > > > > | X-n613-k62      | 613       | 0.92      | 1.02      | 2.34       | 1.39      |
> > > > > > > > | X-n627-k43      | 627       | 0.98      | 1.01      | 2.15       | 1.71      |
> > > > > > > > | X-n641-k35      | 641       | 1.03      | 1.53      | 2.10       | 2.03      |
> > > > > > > > | X-n655-k131     | 655       | 1.07      | 1.13      | 3.63       | 1.63      |
> > > > > > > > | X-n670-k130     | 670       | 1.09      | 1.24      | 4.97       | 1.64      |
> > > > > > > > | X-n685-k75      | 685       | 1.01      | 1.16      | 2.62       | 1.72      |
> > > > > > > > | X-n701-k44      | 701       | 1.03      | 1.16      | 3.04       | 1.51      |
> > > > > > > > | X-n716-k35      | 716       | 1.05      | 1.23      | 3.06       | 1.92      |
> > > > > > > > | X-n733-k159     | 733       | 1.25      | 1.36      | 4.82       | 1.89      |
> > > > > > > > | X-n749-k98      | 749       | 1.61      | 1.55      | 3.42       | 1.95      |
> > > > > > > > | X-n766-k71      | 766       | 1.16      | 1.34      | 4.28       | 2.01      |
> > > > > > > > | X-n783-k48      | 783       | 1.14      | 1.35      | 4.06       | 2.74      |
> > > > > > > > | X-n801-k40      | 801       | 1.46      | 1.63      | 4.12       | 2.47      |
> > > > > > > > | X-n819-k171     | 819       | 1.50      | 1.81      | 4.65       | 2.44      |
> > > > > > > > | X-n837-k142     | 837       | 1.37      | 1.49      | 5.34       | 2.06      |
> > > > > > > > | X-n856-k95      | 856       | 1.30      | 1.40      | 5.27       | 2.10      |
> > > > > > > > | X-n876-k59      | 876       | 1.29      | 1.43      | 5.20       | 2.66      |
> > > > > > > > | X-n957-k87      | 957       | 3.42      | 2.91      | 4.21       | 5.01      |
> > > > > > > > | X-n895-k37      | 895       | 2.64      | 2.58      | 4.23       | 3.91      |
> > > > > > > > | X-n916-k207     | 916       | 3.16      | 3.09      | 8.93       | 6.84      |
> > > > > > > > | X-n936-k151     | 936       | 3.94      | 3.41      | 9.22       | 6.05      |
> > > > > > > > | X-n979-k58      | 979       | 3.03      | 2.95      | 4.22       | 5.06      |
> > > > > > > > | X-n1001-k43     | 1001      | 2.95      | 2.93      | 4.18       | 4.68      |
> > > > > > > > | Li_21           | 561       | 0.76      | 2.12      | 3.55       | 2.07      |
> > > > > > > > | Li_22           | 601       | 0.84      | 1.14      | 3.55       | 2.27      |
> > > > > > > > | Li_23           | 641       | 0.86      | 1.38      | 3.55       | 2.45      |
> > > > > > > > | Li_24           | 721       | 0.99      | 1.58      | 4.05       | 2.20      |
> > > > > > > > | Li_25           | 761       | 1.06      | 1.37      | 4.08       | 2.56      |
> > > > > > > > | Li_26           | 801       | 1.12      | 1.79      | 4.05       | 2.53      |
> > > > > > > > | Li_27           | 841       | 1.64      | 3.12      | 4.06       | 3.76      |
> > > > > > > > | Li_28           | 881       | 1.24      | 2.00      | 4.05       | 2.61      |
> > > > > > > > | Li_29           | 961       | 1.89      | 3.79      | 4.07       | 3.24      |
> > > > > > > > | Li_30           | 1041      | 2.02      | 2.90      | 5.05       | 2.98      |
> > > > > > > > | Li_31           | 1121      | 1.71      | 2.90      | 5.06       | 3.53      |
> > > > > > > > | Li_32           | 1201      | 1.98      | 3.17      | 6.05       | 3.38      |
> > > > > > > > | Leuven1-3001    | 3001      | 9.73      | 9.82      | 68.99      | 15.59     |
> > > > > > > > | Leuven2-4001    | 4001      | 13.45     | 13.68     | 73.96      | 23.73     |
> > > > > > > > | Antwerp1-6001   | 6001      | 12.81     | 12.98     | 596.21     | 24.91     |
> > > > > > > > | Antwerp2-7001   | 7001      | 14.96     | 15.26     | 479.47     | 31.90     |
> > > > > > > > | **Average**     | **1110**  | **2.37** | **2.68** | **28.98** | **4.34** |
> > > > > > > > | Ghent1-10001    | 10001     | 21.41     | 21.66     | -          | 37.26     |
> > > > > > > > | Ghent2-11001    | 11001     | 38.55     | 37.56     | -          | 55.73     |
> > > > > > > > | Brussels1-15001 | 15001     | 131.39    | 139.09    | -          | 166.81    |
> > > > > > > > | Brussels2-16001 | 16001     | 165.94    | 158.71    | -          | 187.48    |

---

> > > > > > > > > ### Comment · Reviewer_cdJ3 · 2022-12-12
> > > > > > > > > **New result**
> > > > > > > > >
> > > > > > > > > Thanks for providing the new results. This really helps to understand the performance of TAM.
> > > > > > > > > Accordingly, I updated my vote.

---

> > ### Comment · Reviewer_cdJ3 · 2022-11-30
> > **CVRP 5000 and CVRP 7000**
> >
> > Thanks for the answers of the questions.
> > One quick questions about the reported time for CVRP 5000 and CVRP 7000. Do you run your algorithm on GPU for the inference? And did you run HGS on CPU?

---

> > > ### Author Response · Authors · 2022-11-30
> > > **Response to CVRP 5000 and CVRP 7000**
> > >
> > > Thanks for your valuable comments.  The answer is yes.  The dividing model of our TAM and AM are run on GPU for inference. The LKH-3 and HGS are run on CPU when used as the solver or second-stage solver.  If our TAM and AM are run on CPU, the inference time will increase. The results are shown in the following table.
> > >
> > > Inference time on CPU for CVRP 5000 and 7000
> > > | Time      | AM     | TAM-AM | LKH3    | TAM-LKH3 |
> > > |-----------|--------|--------|---------|----------|
> > > | CVRP-5000 | 16.18  | 17.08  | 151.64  | 28.54    |
> > > | CVRP-7000 | 30.30  | 31.40  | 501.26  | 48.46    |

---

> > > > ### Comment · Reviewer_cdJ3 · 2022-12-04
> > > > **Thanks**
> > > >
> > > > Thanks for providing the CPU timings. Those still look good.

---

### Author Response · Authors · 2022-11-18
**General response to all reviewers**

We thank the reviewers for their valuable feedback and for seeing merit in our paper. We have tried our best to address your main points. We will also correct all minor issues raised, even if they are not mentioned here.
We first summarize the general themes before answering each reviewer’s specific questions.

**Novelty and Practical motivations.**

One of the biggest obstacles when using the learned VRP heuristics in real-world applications is the generalization ability. Most learned VRP models perform well only on instances with similar number of nodes. TAM bridges the gap by generalizing the model learned on small-scale VRPs to large-scale VRPs. In addition, TAM could easily encode common constraints such as vehicle capacity, maximum vehicle number, and time window by mask function. These advantages make TAM successfully applied in real-world application and obtain superior performance on both synthetic and real-world instances.

TAM is especially motivated by practical online applications. For online applications, such as e-commerce platforms, hundreds and thousands of goods are sold in real-time and then transported to customers with minimum number of vehicles and shortest travel distance. More large-scale VRPs with over 1000 customers need to be solved in seconds to improve logistics or transportation efficiency. Therefore, we need a new VRP solver with the following requirements:
1) Good scalability and generalization-ability for solving large-scale VRPs with limited resources.  To this end, we proposed three generalization techniques to generalize the TAM learned on VRP 100 to solve large-scale VRP with over 5000 nodes while keeping the solution quality better than data-driven heuristics and competitive with traditional heuristics.
2) Fast speed for solving large-scale VRPs in seconds. To this end, we combine the advantages of the learning method and parallel computation for traditional heuristics, which could solve VRP with 5000 nodes in seconds.
3) Flexibility for various practical constraints, such as capacity, time window, maximum vehicle number, limited travel distance or time for each vehicle. To this end, we proposed new rewards and a global mask function to reformulate our original set2set problem as seq2seq problem. Our TAM could then encode various local and global constraints by changing global mask functions without modifying the dividing model.

---

### Decision · Program_Chairs · 2023-01-20

**Decision:**

Accept: poster

**Justification For Why Not Higher Score:**

Might be a niche problem setup for the ICLR community.

**Justification For Why Not Lower Score:**

Reviewers have provided strong support for this work.

**Metareview: Summary, Strengths And Weaknesses:**

The paper presents a new RL algorithm called TAM for  large-scale Vehicle Routing Problems. The reviewers find this work well written, enjoyable, and impressive in terms of its results. There were some concerns about the need for a thorough literature survey, and corresponding expansion of experimental evaluation. However, I believe these concerns have been sufficiently addressed by the rebuttal.

**Note From Pc:**

if the above contains the word "oral" or "spotlight" please see: "oral" presentation means -> notable-top-5% and "spotlight" means -> notable-top-25%. As stated in our emails, we are disassociating presentation type from AC recommendations